# Amazon windthrow disturbances are likely to increase with storm frequency under global warming

Yanlei Feng [1] ✉, Robinson I. Negrón-Juárez[2], David M. Romps[2,3] & Jeffrey Q. Chambers[1,2]

Forest mortality caused by convective storms (windthrow) is a major disturbance in the Amazon. However, the linkage between windthrows at the surface and convective storms in the atmosphere remains unclear. In addition, the current Earth system models (ESMs) lack mechanistic links between convective wind events and tree mortality. Here we find an empirical relationship that maps convective available potential energy, which is well simulated by ESMs, to the spatial pattern of large windthrow events. This relationship builds connections between strong convective storms and forest dynamics in the Amazon. Based on the relationship, our model projects a 51 ± 20% increase in the area favorable to extreme storms, and a 43 ± 17% increase in windthrow density within the Amazon by the end of this century under the high-emission scenario (SSP 585). These results indicate significant changes in tropical forest composition and carbon cycle dynamics under climate change.

Convective storms account for ~50% of the tree mortality across Amazonia at scales ranging from individual trees[1] to large forest gaps exceeding thousands of hectares[2–5], resulting in uprooted and broken trees (windthrow[6]). Despite its importance, a detailed understanding of the mechanisms of convective storms producing windthrow events remains unclear. Given the sensitivity of convective storms to climate change, a better understanding of the interactions between convective storms and windthrow events is needed to explain changes in forest composition and ecosystem processes under the elevated disturbance regimes expected with continued climate system warming.

Convective storms are deep moist convective cells, formally known as Mesoscale Convective Systems (MCSs). In the Amazon basin, 64% of MCSs last 3–5.5 hours, while 0.7% of MCSs have a lifespan equal to or over 24 hours[7]. Some of the most extreme convective systems, such as squall lines, also occur over the Amazon region[8]. The occurrence of MCS is most common between afternoon and evening in the local time[9,10]. The development of these convective cells includes an updraft of moist air, which forms the cumulonimbus cloud, and a downdraft, which is associated with heavy precipitation, strong surface winds, hail, and cloud-to-ground lightning[9,11]. Warm and moist near-surface air in the Amazon region provides a favorable environment for the development of MCSs[7]. MCSs are driven by moist convective instability, which can be well represented by convective available potential energy (CAPE). CAPE determines the speed limit of upward convection. CAPE is also fundamental to our understanding of future convective storms[12]. CAPE increases robustly with warming across the tropics in ESMs, suggesting an increasing likelihood of favorable environments for most severe storms[11,13–15]. In addition to CAPE, vertical windshear physically displaces downdraft air from the storm, which prolongs and intensifies storms, and also further influences the development of further convections[11,16].

In the Amazon, MCSs have significant impacts on forest ecosystem processes, community structure and composition, and the regional carbon balance[5,17–19]. MCSs produce 50–90% of the annual precipitation in the tropics, shaping key aspects of tropical forest function and structure[11]. Severe winds and heavy rainfall associated with MCSs also cause widespread windthrows in the Amazon rainforest[10,20]. At the continental scale, Nelson et al.[5] mapped 330 large windthrows in the Brazilian Amazon using satellite images and explored the size and spatial pattern of windthrow events. Field studies

[1]Department of Geography, University of California, Berkeley CA, USA. [2]Climate and Ecosystem Sciences Division, Lawrence Berkeley National Laboratory, Berkeley CA, USA. [3]Department of Earth and Planetary Science, University of California, Berkeley CA, USA. ✉e-mail: ylfeng@berkeley.edu

of windthrows in the Amazon show there is a positive correlation between the satellite-derived disturbance intensity metrics and field-measured tree mortality within windthrows[2,17]. The study of a windthrow event in the Manaus region documented a high regional mortality from a single squall line[17]. In addition to wind and rainfall, lightning associated with convective storms can also be a major agent of canopy disturbance and large tree mortality[21,22].

Previous estimates of windthrow events in the Amazon relied on measuring extreme rainfall events and cloud top temperature. The highest frequencies of windthrows were found to coincide with the frequency of heaviest rainfall days[2,5,23] and strong convective activities measured from cloud top temperature[10,18]. Storms that caused windthrows were the most intense storms, capable of ascending furthest into the upper troposphere or lower stratosphere; therefore, these storms come with the lowest cloud top temperature and heaviest rainfall[8,10,24,25]. However, extreme rainfall events are poorly projected over the tropical regions in ESMs[26], and realistic storm cloud top projections are not represented in the models. Moreover, it is still unclear how the cloud physics of MCSs will change under a warming climate. Therefore, a reliable climate variable with robust model projection in the future that correlates with windthrows is lacking, and the current suite of ESMs[27] is unable to project future forest windthrows dynamics.

This study seeks to investigate the spatial pattern of windthrow events that are larger than 25,000 m² (hereafter 'large windthrows') and their correlation with CAPE over 30 years while demonstrating that CAPE is an appropriate proxy for estimating the spatial pattern of large windthrow events in the Amazon. CAPE has been well studied in ESMs[13,14,16,28]. Compared to the relatively uniform geographic variability of weak windshear[11] over the Amazon region (Supplementary Fig.1), CAPE shows a greater spatial variation. Furthermore, the frequency of severe thunderstorm environments increases with warming across the tropics, mainly due to changes in the distribution of CAPE[13]. Therefore, CAPE is selected for this study as the only climate variable for the spatial pattern analysis of windthrows. A model is also developed using CAPE to project future windthrow events, and it projects a large increase of windthrow events in the Amazon region by the end of this century.

In this study, we manually map 1012 large windthrow events encompassing 30 years from 1990–2019 (Fig. 1a) using images from Landsat 8 OLI/TIRS collection (USGS) over the entire Amazon region (see Methods for the details). Windthrow density is generated using all of these windthrow events. This study uses only large windthrow events that can be identified using Landsat satellite images. These large windthrows are produced by extreme convective storms[10], such as squall lines[17]. Although this approach omits the vast majority of small windthrow events (≤25,000 m²)[19], the general spatial pattern of large windthrows occurrence is used as a tractable proxy for all windthrows. Hourly CAPE is obtained from ERA5 reanalysis data[29]. The means of CAPE in the local time 13:00–19:00 (UTC 17:00–23:00) over 1990–2019 are calculated (hereafter mean afternoon CAPE).

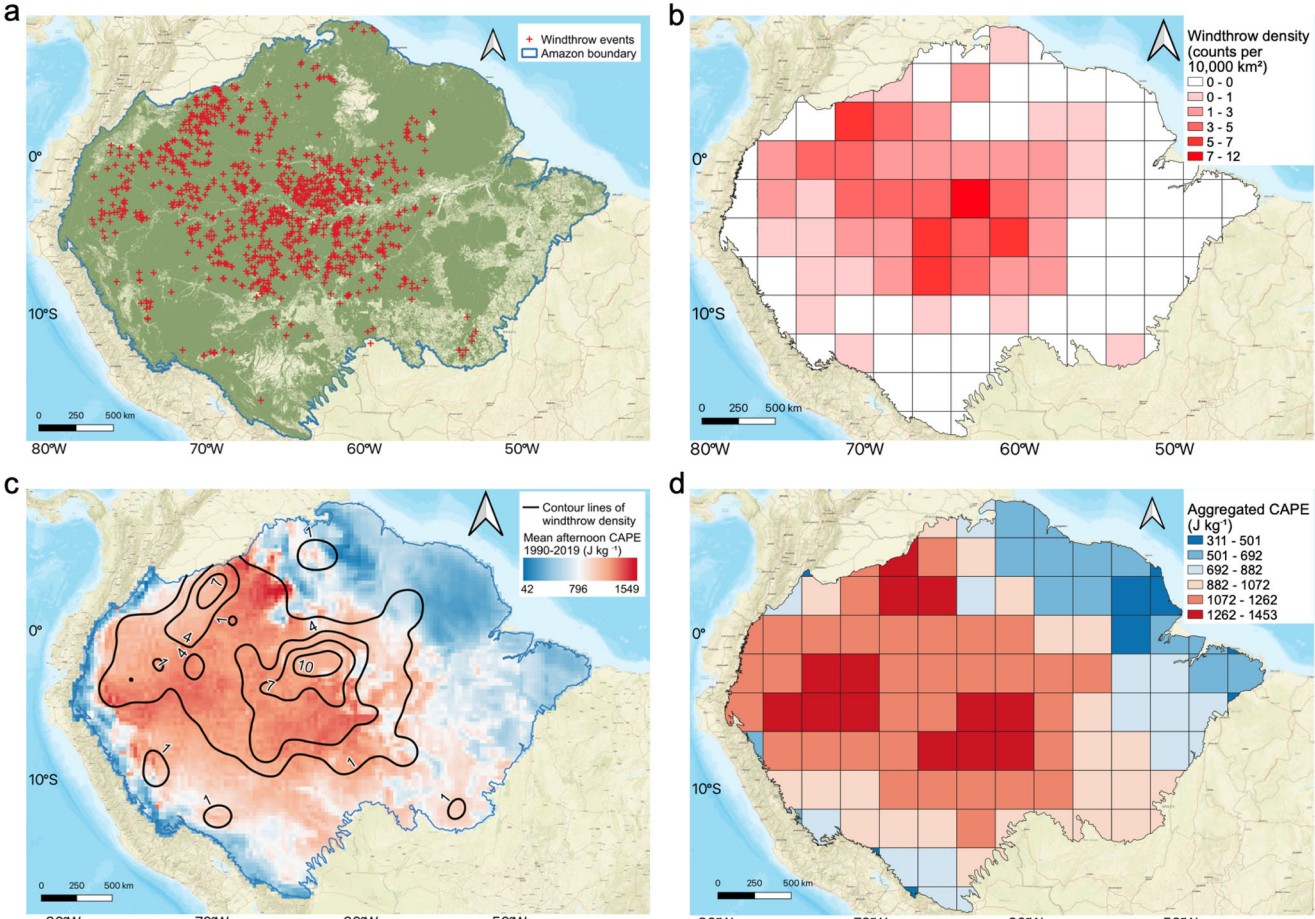

**Fig. 1 | The spatial pattern of windthrows and mean afternoon convective available potential energy (CAPE). a** 1012 Windthrow events identified manually using Landsat 8 images, green color in the background represents forested area. **b** Windthrow density in 2.5° × 2.5° grids. **c** Contour lines of windthrow density (counts per 10,000 km²) over the mean afternoon CAPE at 0.25° resolution. **d** Mean afternoon CAPE aggregated in 2.5° × 2.5° grids using the 90th percentile[13] over the grid.

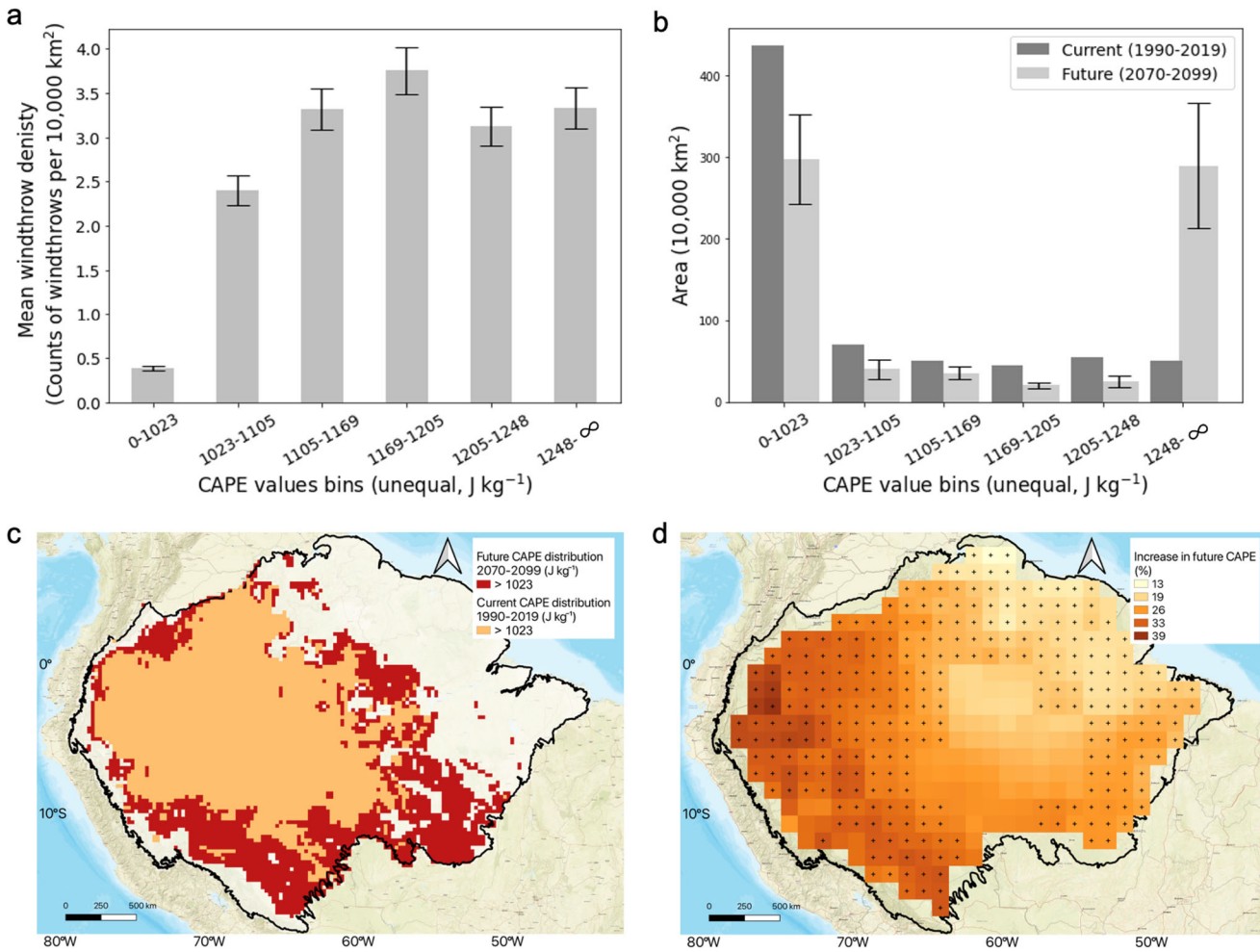

**Fig. 2 | The relationship maps convective available potential energy (CAPE) to windthrow density and future increase in CAPE simulated by Earth system models under the high-emission scenario. a** Mean windthrow density as a function of CAPE values, calculated using the data shown in Figs. 1a, c. The boundaries of the CAPE bins were selected to have the same number of observed windthrows in each bin to avoid noise at the tails. The error bars (SD) of the windthrow density were generated using 10,000 bootstrapped samples of the 1012 windthrow points. The lower and upper CAPE bin boundaries were expanded to a minimum of 0 and a maximum of infinity with an assumption that the windthrow density is similar for the neighboring CAPE values. **b** The area of the Amazon region in each CAPE bin for the past 30 years and for the last 30 years of the century. The

error bars (SD) of future CAPE were generated using scaled 2070–2099 CMIP6 CAPE from 10 ESMs. **c** The increase in area with CAPE over 1023 J kg$^{-1}$, with orange pixels representing mean 1990–2019 ERA 5 CAPE higher than 1023 J kg$^{-1}$ and red pixels representing mean scaled 2070–2099 CMIP6 CAPE higher than 1023 J kg$^{-1}$. **d** Ensemble-mean increase of CAPE from the current climate (1990–2014) to the future climate (2070–2099) under the SSP585 scenario. Since CMIP6 models provide historic simulations only up to 2015, data from 2015 to 2020 are not included. Stippling indicates regions where all 10 ESMs agree on the increase of CAPE, with CAPE calculated using daily surface pressure and atmospheric profiles at standard pressure levels.

## Results

### The spatial pattern of windthrows and CAPE

As seen in Fig. 1, the spatial pattern of mean afternoon CAPE matches with the density of large windthrows. High densities of large windthrow events are in the northwest and central Amazon regions (Fig. 1b) that experience the highest CAPE (Fig. 1d). Few large windthrow events are found on the northeast side of the Amazon region, where CAPE is low, indicating less severe storms[30]. The region with the highest windthrow density over 12 events per 10,000 km$^2$ is in the state of Amazonas in Brazil, in between Rio Negro and the Amazon River. Another dense region with windthrow density over 7 events per 10,000 km$^2$ is near the east border of Columbia. Both regions occur in locations where the mean afternoon CAPE exceeds 1000 J kg$^{-1}$ (Fig. 1c). In addition to mean afternoon CAPE, mean daily CAPE over 30 years have a similar spatial pattern (Supplementary Fig. 2), but it exhibits high values over rivers due to the nighttime CAPE intensification (Supplementary Fig. 3).

### A relationship maps CAPE to windthrow density

We found an empirical relationship using mean afternoon CAPE to reproduce the spatial variability of windthrows and project the frequency of future windthrows in the Amazon (Fig. 2a). The relationship uses the locations of windthrow events shown in Fig. 1a and the distribution of mean afternoon CAPE shown in Fig. 1c. The values of CAPE at the locations of the 1012 windthrow events are sorted and split into six equal-sized groups (of 168 or 169 each). This determines the boundaries of the CAPE bins used in Fig. 2a, which plots the current windthrow density, i.e., the number of windthrow events (168 or 169) divided by the area of the Amazon with that CAPE (see Methods for the details). Fig 2a shows that a CAPE threshold at 1023 J kg$^{-1}$ can be used to define if the environment is favorable for high density of windthrows. A similar threshold is defined by a mechanistically-based approach using the data shown in Fig. 1b, d (Supplementary Fig. 4) and is supported by convection theory, numerical models, and observations in previous studies[31,32]. 62% of the Amazon area has a relatively low windthrow density where CAPE is lower than 1023 J kg$^{-1}$. 38% of the Amazon region

**Table 1 | Future changes predicted by Earth system models (ESMs) under the highest emission scenario SSP585**

| ESMs | Increase in area with CAPE > 1023 J kg$^{-1}$ (%) | Increase in wind-throw density (%) |
|---|---|---|
| BCC-CSM2-MR | 40 | 32 |
| CanESM5 | 60 | 12 |
| FGOALS-g3 | 13 | 50 |
| MIROC6 | 70 | 58 |
| MIROC-ES2L | 66 | 56 |
| KIOST | 40 | 33 |
| MRI-ESM2-0 | 71 | 60 |
| ACCESS-CM2 | 64 | 52 |
| MPI-ESM1-2 | 27 | 22 |
| CNRM-CM6-1 | 61 | 51 |
| Ensemble Mean | 51 | 43 |
| One standard deviation | 20 | 17 |

Increase in area with storm-favorable environments was calculated as the percentage of the increased area with storm-favorable environments from the current ERA 5 CAPE (1990–2019) to scaled CMIP6 future CAPE (2070–2099) over the Amazon region. Projected windthrow density increase was calculated using the model developed in this study. All numbers are rounded to the nearest integer. Computation details can be found in the methodology.

has a mean CAPE over 1023 J kg$^{-1}$ (Fig. 2c), which provides a storm-favorable condition with high windthrow density. In general, windthrows occur nearly 8 times more frequently in the favorable environments (CAPE > 1023 J kg$^{-1}$, hereafter storm-favorable environments) than in the non-favorable environments (CAPE ≤ 1023 J kg$^{-1}$). The mean uncertainty (SD) of these values, estimated via bootstrapping, is 0.2. The relationship shows that CAPE is a strong predictor of the spatial pattern of large windthrow events in the Amazon region and provides a step function that can model windthrow density using CAPE values.

**Future increase in storm-favorable environments and windthrows**

We developed a model based on the CAPE and windthrow density relationship to assess how climate change will affect CAPE and then affect the density of future large windthrows in the Amazon region. Outputs from 10 ESMs in CMIP6 were analyzed and incorporated into our model. Figure 2b shows the distribution of area by the current and future CAPE values over the Amazon region. The comparison between the spatial pattern of current and future CAPE can be found in Supplementary Fig. 5a, b. The current CAPE distribution was generated from mean afternoon ERA5 CAPE. For future CAPE in 2070–2099, we calculated the mean of CAPE values from 10 CMIP6 ESMs and adjusted the values to make them comparable with ERA 5 CAPE values (see Methods for the details). Table 1 lists the fractional changes in storm-favorable CAPE for the ESMs in our ensemble (see Supplementary Table 1 for details of the ESMs) under the highest emission scenario SSP585. All 10 ESMs agree on an increasing trend of CAPE over the Amazon region, with a mean increase of 26 ± 9% (mean ± SD) from the past 30 years (1990–2015) to the last 30 years of the 21st century (Fig. 2d). There is a high level of agreement between models on the spatial pattern and magnitude of the CAPE increase (Fig. 2d). The magnitude of future CAPE indicates strong atmospheric instability that could produce more frequent storms. While the fraction of area that is storm-favorable in the current mean CAPE map is 38%, ESMs project that 58 ± 8% of the Amazon region becomes storm-favorable by the end of the 21st century. The area with CAPE over 1023 J kg$^{-1}$ increases 51 ± 20% by the end of the 21st century (Table 1), indicating a much larger area with storm-favorable conditions within the Amazon. An area of ~1,390,000 km$^2$ in the northwestern, southern, and central Amazon, including forested areas in central Columbia, the northern part of Bolivia, the south-eastern part of Peru, and central Brazil, where

windthrows are infrequent now, are projected to experience an eightfold or greater increase in windthrow events by 2100 (Fig. 2c). The northwestern and western Amazon, where large windthrow density is relatively high now (2.4 windthrows/10,000 km$^2$), are estimated to experience a 33–50% increase in the number of windthrow events in the next 50–80 years (Supplementary Fig. 5c).

Using the model, all ESMs in our ensemble predict a significant increase in future windthrow events. Overall, windthrow density in the Amazon region is estimated to increase 43 ± 17% over the 21st century (Table 1).

## Discussion

Tree mortality plays important roles in determining forest carbon balance across Amazonia, and variable disturbance regimes increase uncertainty in the tropical forest carbon sink capacity over time. Extreme convective storms are important drivers of tree mortality in the Amazon region and affect the biomass patterns and function composition of Amazonian forests[33]. In this study, we provide a framework for representing the coupling between forest mortality on the land surface and windstorms in the atmosphere. Previous studies showed that tree mortality in the central Amazon was higher in wet months as a result of extreme storms, even in drought years[34,35]. Elevated forest mortality related to extreme storms in the central Amazon was found in the La Niña wet year of 1990, La Niña year of 1999 followed by prolonged warmer temperatures and drought in the previous years, and in the drought year of 2005[34]. The relationship between extreme storms and tree mortality also implies that increasing frequency of convective storms[36] may contribute to the observed increase in tree mortality and a weakening of the carbon sink across Amazonia[37–39].

This analysis highlights the potential for predicting the rate of future windstorm-driven tree mortality, a driver of tree mortality that currently is not included in ESMs, and emphasizes the need to improve land-atmosphere coupling in ESMs. The results in this paper generated using ESM outputs and the model indicate that over half of the Amazon region is projected to experience average conditions favorable for intense convective storms with a large increase in windthrow events by the end of this century under the highest emission scenario SSP585. The projected increase in the number of large windthrows potentially facilitate the establishment of forest species with fast turnover rate[2,33,34]. These species are vulnerable to disturbances, which leads to greater forest mortality and in turn create more forest gaps. This positive feedback loop speeds up forest dynamics with faster carbon-water and nutrient cycling and less carbon storage[2,33,34]. The projected significant increase in tree mortality associated with windthrow density in the Amazon by the end of the 21st century has the potential to re-shape tropical forest structure and composition, with sizable impacts on regional carbon balance.

This first step in projecting the impact of future extreme wind events on Amazon forests has several caveats, which we discuss here to facilitate the interpretation of our results. Potential biases can exist in disturbance identification and sampling effort. 2.8% of the Amazon region was not included in this study due to the lack of clear remote sensing images (clouds <20%). Windthrows without a characteristic shape were eliminated from this study. We only considered the spatial pattern of large windthrows in this study while omitting the vast majority of windthrows that are smaller than 25,000 m$^2$. Since the focus of this study is on the spatial pattern of large windthrow events, the underestimation of windthrow events will not have a significant impact on the results. In addition, the large windthrow dataset in this study provides a comprehensive view over the entire Amazon which would be difficult to achieve using small-scale windthrow datasets.

There are also biases arising from regional variation in vegetation composition or vegetation dynamics. Compared with the central Amazon, forests in the northwestern Amazon are more vulnerable to

windthrow mortality and recover faster due to a number of factors, including tree stem density, soil type, root architecture, tree diversity, microclimate condition, topographic exposure[2,40]. A previous study showed that windthrows were spatially more common in the north-western Amazon than in the central Amazon[2]. With high windthrow tree mortality, forest in the northwestern amazon took ~20 years to recover to 90% of pre-disturbance biomass, and forests in the central Amazon took ~40 years to recover, and the biomass recovery depends on the windthrow severity and time since disturbance[33,40]. Therefore, fewer large windthrows may be identified in this study in the north-western Amazon due to relatively faster recovery. More accurate windthrow identification could be done on a yearly basis. Such analysis, however, is beyond the scope of the current study.

The function of CAPE and large windthrow density developed in this study is based on the assumption that increases in CAPE will result in increases in the frequency of extreme convective storms, which create more large windthrows. This assumption is supported by convection-permitting models that project an increase in extreme storm frequency with a warming climate[41]. However, the assumption may not be applied to weak or moderate convective storms, which are projected to decrease in the future[41]. Studies on small windthrows (<25,000 m$^2$) and the corresponding convective storms can be carried out to explore their relationships. Future studies could involve convection-permitting models[42] that are able to simulate MSCs to reduce the uncertainty in storm process behavior when exploring changes in storm frequency and intensity in the Amazon region.

The study refrains from discussing the relationship between storm intensification and forest structure changes. How the increasing CAPE in the Amazon region affects precipitation rate[43] and wind gust potential[44], which are the major causes of tree mortality[10,20], has not been well explored. We emphasize the need for additional field observation datasets of severe convective winds and extreme rainfall rate associated with convective storms in the Amazon region. Such observations would improve the understanding of how forest dynamics link with convective storm intensity.

## Methods

### Identification of windthrow events

Landsat images from January 1st 2018 to December 31st 2019 were filtered on 20% or less of cloud coverage, and only the least cloudy image at each location was selected to make an image composite covering the entire Amazon region. In total, 395 least cloudy Landsat 8 images within the Amazon boundary during 2018–2019 were selected and displayed in false color (red: shortwave infrared band, green: near-infrared band, blue: red band) on Google Earth Engine for windthrow events identification (Supplementary Fig. 6). Hollow regions on Supplementary Fig. 6 (2.8% of the total area of the Amazon region) indicated that no clear images with <20% cloud were found in these areas. Spectral mixture analysis was applied on each image to detect potential windthrows (similar methods have been used in previous windthrow studies[5,17]). Each pixel was unmixed to fractions of image-derived endmembers, including green vegetation (GV), non-photosynthetic vegetation (NPV), and shade. GV and NPV fraction images were normalized without shade and then used to identify windthrows. Windthrows were identified manually as large fan-shape areas with high NPV fraction. Each potential windthrow was then visually checked using false color images and evaluated based on authors' 15-year experience working with windthrow and remote sensing[2,3,17,19]. "New" windthrows that occurred within 1 year were spectrally more visible based on their clear fan-shape[5,10] (diverging from a central area with small pixels scattered at the tail) and their relatively distinguishable reddish colors (Supplementary Fig. 7a, due to high reflectance in shortwave infrared band from woody biomass), while "old" windthrows (Supplementary Fig. 7b) occurred >1 year before the identification were displayed in bright green colors (due to

reflectance in near-infrared band from the pioneer species). "Old" windthrows account for ~80% of total identified windthrows, and they were verified using historical Landsat images that can go as far as 1984 (when Landsat 5 was launched). "Old" windthrows were validated once they were found with clear shape and more distinguish color on the historical Landsat images (Supplementary Fig. 7c). 10–15% of "old" windthrows without fan-shape were eliminated from this study because it was hard to identify if they were windthrows or other types of forest disturbance. The minimum size of windthrows identified in this study was 25,000 m$^2$. This process generated the location and rough size of 1012 visible (both "old" and "new") windthrow scars with fan-shaped patch, scattered small disturbance pixels tails, and an area of over 25,000 m$^2$ (Supplementary Fig. 8). Based on a gap-size probability distribution function that simulates the entire disturbance gradient from all sizes of windthrows[19], the proportion of total tree mortality represented by large windthrows (>25,000 m$^2$) identified in this study is 0.5–1.1%.

Among 1012 visual identified windthrows, the occurrence year of 125 windthrows were identified using Landsat 5,7,8, MODIS, and TRMM dataset (Supplementary Table 2), and 38 windthrows from these 125 windthrows had clear remote sensing evidence to validate their occurring date (Supplementary Table 3). It is difficult to get the accurate year and date of occurrence of all identified windthrows. Previous studies showed that windthrows in the northwestern Amazon took ~20 years to recover to 90% of "pre-disturbance" biomass from all damage classes while forests in the central Amazon took ~40 years to recover[40]. The biomass recovery depends on the windthrow severity and time since disturbance[33]. Based on the recovery time (20–40 years) and the time of windthrow identification (2018–2019), we esti-mated that these 1012 windthrows most likely occurred within 30 years (between 1990 and 2019), and the estimated occurrence period was validated using the range of the occurrence year (1986–2019) of 125 windthrow cases.

### Windthrow density data

The windthrow density shown in Fig. 1b was generated using 1012 windthrow points in QGIS[45]. We created a 2.5° by 2.5° grid map, and the windthrow density was calculated by counting the number of wind-throws in each grid. These values were then converted to a density with units of counts of windthrows per 10,000 km$^2$. We chose 2.5 degrees to aggregate the data to make sure that over 50% of grids have at least 1 windthrow event while still preserving the spatial distribution of mean afternoon CAPE over the Amazon. The contour lines displayed in Fig. 1c were generated using the "Contour" function on the windthrow density map in QGIS.

### Meteorological data

To derive the correlation between windthrow density and meteor-ological variables, we used ERA 5 global reanalysis hourly CAPE on single levels from 1979 to present at 0.25° × 0.25° resolution provided by the European Center for Medium-Range Weather Forecasts. ERA 5 CAPE was computed by considering parcels of air departing at differ-ent pressure levels below the 35 kPa level, with maximum–unstable algorithm under a pseudo-adiabatic assumption[46]. Afternoon mean CAPE map was calculated as the average of hourly CAPE data from 17:00–23:00 UTC (13:00–19:00 local time in Amazon) over all the months between 1990 and 2019. We chose to average CAPE over 30 years because these windthrow events occurred in these 30 years and calculating the average can help capture the overall spatial pattern of CAPE and minimize the influence of interannual climate variability on windthrow events.

To project future windthrow density in the Amazon for the end of the 21st century, we analyzed meteorological output from 10 ESMs that participated in CMIP6 (https://www.wcrp-climate.org/wgcm-cmip/wgcm-cmip6). The models used in this research were listed in

Supplementary Table 1. We extracted daily surface temperature (tas), specific humidity (huss), surface pressure (ps), temperature (ta) from these models to calculate daily nondilute, near-surface-based, adiabatic CAPE. CMIP6 CAPE was calculated by considering the buoyancy of a near-surface parcel lifted adiabatically to a series of discrete pressure levels (100 kPa to 10 kPa in increments of 10 kPa). CMIP6 CAPE is calculated as follows:

$$CAPE = \sum_{i=1}^{10} \mathrm{d}p \mathrm{H}(b_i) b_i \qquad (1)$$

Where $\mathrm{d}p = 10$ kPa, H is the Heaviside unit step function, and $b_i = \frac{1}{\rho_i} - \frac{1}{\rho_{e,i}}$, with $\rho_i$ being the parcel density at pressure level $i$ and $\rho_{e,i}$ being the environmental density at pressure level $i$.

The future projections in our analysis were based on SSP585, a high-emission scenario with high radiative forcing by the end of the century. We calculated mean daily CAPE over 1990–2015 as current CMIP6 CAPE and mean daily CAPE over 2070–2099 as future CMIP6 CAPE. Since different approaches were used to calculate ERA 5 CAPE and CMIP6 CAPE[47], the absolute CAPE values of the two datasets are not comparable. Therefore, for each ESM model, we scaled future CMIP6 CAPE by multiplying, grid-wise, the delta CAPE generated from an individual model in CMIP6 with the ERA 5 current mean afternoon CAPE (Fig. 1c) as follows:

$$delta\,CAPE = (CAPE_{CMIP6\_future} - CAPE_{CMIP6\_current})/CAPE_{CMIP6\_current} \qquad (2)$$

$$CAPE_{scaled\_CMIP6\_future} = (1 + delta\,CAPE) \times CAPE_{ERA5-current} \qquad (3)$$

The delta CAPE indicated the projected increase in CAPE from 1990–2015 to the end of the 21st century. In this way, a scaled CMIP6 future CAPE map was generated for each model, and an ensemble-mean scaled CMIP6 CAPE map over 10 ESM models can be found in Supplementary Fig. 5b. The scaled CMIP6 future CAPE values were within plausible range compared to the ERA 5 current mean afternoon CAPE values, and both current and future CAPE maps were used to produce the increase in area with high CAPE values ($>1023$ J kg$^{-1}$) in Table 1. However, it is worth noting that the scaling with relative changes in delta CAPE (%) is more sensitive to CMIP historical baseline conditions than absolute changes of CAPE (J kg$^{-1}$), which will likely introduce a larger scaled spread (min/max CAPE changes).

The increase in area with storm-favorable environments was calculated as follows:

$$Increase = (area_{future} - area_{current})/area_{current} \qquad (4)$$

Where area$_{current}$ is the area of CAPE $> 1023$ J kg$^{-1}$ for current ERA 5 CAPE, and area$_{future}$ is the area of CAPE $> 1023$ J kg$^{-1}$ for the scaled CMIP6 future CAPE.

### A model of windthrow density
We developed a model based on the relationship between satellite-derived windthrow density and mean afternoon CAPE from the ERA 5 reanalysis over 1990–2019. The non-parametric model provides a look-up table of windthrow density as a function of CAPE within the range of observations. Counts of observed windthrow events and Amazon's area were separately binned by CAPE using the same bins, producing two histograms of CAPE. The ratio of the former to the latter gives the density of windthrow events (windthrow events per area) as a function of CAPE. To avoid noise at the tails of the histograms, the six CAPE bins were chosen such that each bin would have about the same number of windthrow events (either 168 or 169). The total number of windthrow events is given by the sum over

bins of the product of windthrow density and area. The minimum and maximum of current ERA 5 mean afternoon CAPE was 42 and 1549. The minimum CAPE value of the first bin was extended to 0 and the maximum CAPE value of the last bin was extended to infinity under the assumption that the windthrow density is similar for neighboring values. Based on the windthrow density and CAPE relationship used in the model, it is the increase in the area with high CAPE that then leads to an increase in the number of windthrow events.

It is worth noting that the future windthrow density produced by models may be underestimated because the windthrow observations within regions with high CAPE were incomplete due to high cloud coverage. Moreover, the non-parametric model makes the conservative assumption that the windthrow density does not increase at higher, as-yet unobserved values of mean afternoon CAPE.

### Future projections of windthrow density
We combined the non-parametric relationship (Fig. 2a) with the future CAPE map generated from the ten CMIP6 ESMs (adjusted by ERA 5 mean CAPE values) to estimate the changes in windthrow density at the end of the 21st century. We estimated uncertainties for windthrow density projections by combining information about model-to-model differences. The analysis yielded a set of 10 estimates. The overall windthrow density increase and uncertainty were estimated using the mean increase and one standard deviation from the ensemble of the 10 models.

## Data availability
The windthrow location datasets are available from the corresponding authors upon request. Current CAPE (1990–2019) datasets were retrieved from ERA5 hourly data on single levels from 1959 to present[48] (available at https://cds.climate.copernicus.eu/cdsapp#!/dataset/reanalysis-era5-single-levels?tab=overview). Future CAPE datasets were calculated using daily surface temperature (tas), specific humidity (huss), surface pressure (ps), temperature (ta) variables from 10 models in CMIP6 (available at https://esgf-node.llnl.gov/search/cmip6/). Source datasets, including windthrow density datasets, processed mean ERA 5 CAPE datasets, and processed mean scaled CMIP6 future CAPE, and processed datasets to produce Fig. 2a and future windthrows are available at https://doi.org/10.15486/ngt/1883604[49].

## Code availability
The codes used in generating this paper's results are available at https://doi.org/10.15486/ngt/1883604.

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

## Acknowledgements

This research was supported as part of the Next Generation Ecosystem Experiments-Tropics, funded by the U.S. Department of Energy, Office of Science, Office of Biological and Environmental Research under contract number DE-AC02-05CH11231. We acknowledge the World Climate Research Program, which, through its Working Group on Coupled Modeling, coordinated and promoted CMIP6. We thank the climate modeling groups for producing and making available their model output, the Earth System Grid Federation (ESGF) for archiving the data and providing access, and the multiple funding agencies that support CMIP6 and ESGF.

## Author contributions

R.I.N. and J.Q.C. conceived the study. Y.F. and D.M.R. designed the research and developed the methods. R.I.N. collected the datasets of 1012 large windthrows. Y.F. and D.M.R. collected and calculated CAPE datasets from ERA 5 and CMIP6. Y.F. performed the analyses and wrote the manuscript. R.I.N., J.Q.C., D.M.R., and Y.F. have reviewed and helped to revise the manuscript.

## Competing interests

The authors declare no competing interests.
