## [Peer Review File · Nature Communications]

Amazon windthrow disturbances are likely to increase with storm frequency under global warmingREVIEWER COMMENTS

Reviewer #1 (Remarks to the Author):

This study quantifies the relationship between large windthrow events, as detected visually via Landsat images, and a proxy for convective storm activity (CAPE) across the Amazon basin. Aggregating data at the 2.5 degree scale revealed a strong relationship between CAPE and windthrow density across the Amazon basin for the last 30 years. This spatial relationship is then combined with future climate models to predict how windthrow events will change across the next century as a function of changes in CAPE. The estimates vary, but all show a substantial increase in windthrow frequency that has major consequences for future forests. The study is well written and succinctly addresses a major knowledge gap. Overall, the results are very important and have the potential to meaningfully improve how scientists think about the ways that changing climate affects tropical forests. In particular, this work is meaningful because it highlights an understudied aspect of global change that could play a dominant role in shaping future forests. However, I do have some major comments and concerns related to various parts of the manuscript.

Major comments

1. There is substantial potential for bias based on the current description of the methods for identifying windthrows. Concerns related to these potential biases should be addressed, potentially through a combination of clarifications to the methods and supporting analyses. Based on the description provided in the methods (lines 256-268), it is not clear how the authors handled biases in (1) the spatial distribution of image selection across the region (i.e., were the 395 images used in this study evenly distributed across the Amazon? Or were some areas sampled more than others?); (2) temporal differences in image selection (for example, if two regions have 6 images each, were those images distributed evenly across 30 years in one region but clustered in the same 6-year period in the second region?); (3) biases in visual disturbance identification and sampling effort among images (i.e., were all images visually scanned for the same amount of time? Were they scanned until new windthrow detection rates dropped below a certain level? Or standardized in some other way?); or (4) biases arising from regional variation in vegetation composition or vegetation dynamics (e.g., high productivity obscuring disturbances more rapidly). All four of these factors could influence spatial variation in the disturbance density metrics used in this study. Explicitly state and, when possible, quantify the potential biases associated with each of these three issues. If biases are identified, it will be important to estimate and correct for these biases.

2. The linear regression expressed in Fig. 2d is not appropriate for the relationship between windthrow density and CAPE. Indeed, the current relationship predicts impossible negative values for all CAPE under 400 J kg⁻¹. Windthrow density cannot possibly be negative, so the fitted relationship should approach an asymptote at 0 on the y-axis (or the intercept should be fixed at 0, but that doesn't seem as though it would produce a good fit). Please provide a more realistic form for the relationship between these two variables. This is particularly important because the strength of this linear relationship and its future projections are the two primary results of this study.

3. Be more transparent about what is being measured in this study and its implications. The manuscript consistently implies that this study measures the relationship between convective storms and total windthrow rates by omitting information about the windthrow events being measured. Specifically, it is not explicitly stated that all relationships focus on uncommonly large windthrow events that make minor contributions to total tree mortality and total windthrow rates, as shown by the authors' previous work (Chambers et al. 2013). It is likely that the authors chose to focus only on large windthrow events because this is feasible, and that decision is reasonable. However, it should be explicitly stated in the introductory text (potentially in the paragraph on lines 72-81) that this study uses large windthrow events as a tractable proxy for all windthrow. Similarly, the abstract and results should explicitly state that

his study shows relationships between large windthrow events and CAPE, not 'windthrow' without qualification. Somewhere early in the text it should be stated that this approach omits the vast majority of windthrow events (Fig. 2 of Chambers et al. 2013). Lastly, based on the authors' previous work, it should be possible to approximate the proportion of total windthrow events (or the proportion of total disturbed area) represented by the windthrow events included in this study. Even if it was imprecise, such an estimate would give the reader clarity about the detection efficiency of windthrow events used in this study.

4. Similar to point 3, some statements about the importance and novelty of this work are a little misleading. This study is meaningful on its own, and these overstatements unnecessarily undermine the important findings. I highlight two examples here, but it would be good to be more cautious throughout the text.

First, the opening sentence states that windthrows cause more than 50% of tree mortality in Amazonia. The study being cited (Esquivel-Muelbert et al. 2020) estimates breakages and uprooting as 51% of mortality across Amazonian forest plots with a 95% confidence interval of 48-54%. I doubt anyone reading 'more than 50%' interprets this to mean that the chances of being at or below 50% are as nearly as high as above 50%. This overstatement is unnecessary.

Second, the abstract states that, "This relationship builds connections between strong convective storms and forest dynamics for the first time in the Amazon." This is not true. Previous work showed that lightning frequency (also associated with convective storms, as demonstrated by decades of research on thunderstorms and by Dr. Romps previous work on CAPE and lightning; Romps et al. 2014) was strongly associated with forest dynamics across the Amazon, in terms of fewer large trees, high rates of woody biomass loss, and less aboveground biomass in forests with higher lightning frequency (Gora et al. 2020). Narrowing the focus on the patterns analyzed in this study would avoid overstating the findings.

5. The methods for image selection (lines 258-268) were not detailed enough to facilitate reproducibility. There were a number of issues, listed with letters below:

a. Since 30x30m is the size of a Landsat pixel, the description of windthrow selection suggests that any event with at least one pixel could be included. However, that doesn't seem to be the case based on the low number of windthrow events. Clarify the minimum size threshold used here.

b. Referring to Fig. S6 for clarification, it is not clear whether the disturbances shown in each image were considered one windthrow event or how this approach would differentiate nearby windthrow events from each other. In general, there is a need for more detail about how windthrow events were classified and how they were separated from each other and from other disturbance events.

c. As for assigning the date of occurrence, it is not clear how the date was determined from the text or why the date of occurrence for 125 disturbances indicated that all 1012 disturbances occurred between 1990-2019. If these 125 dated disturbances were used to identify similarly aged disturbances, then provide details about how many disturbances were removed because of an age mismatch. If none were removed, explain why this method should be viewed as effective given the probability that a random sample of images would include disturbances from a prior period.

d. There may be a word missing from the sentence on lines 258-261 (potentially an "and" before "an area over"?), and as a result, it is not clear if all 1012 windthrow events have all three characteristics described in this sentence.

6. The study should acknowledge the importance of lightning as an agent of forest disturbance associated with convective storms. Lightning is mentioned once, but its role as an agent of disturbance is ignored. Lightning could play a key role in the effects of changing convective storm frequency on future forests and acknowledging that

consequence would strengthen this study. Specifically, lightning was recently shown to be a major agent of canopy disturbance and large tree mortality (Yanoviak et al. 2020; Gora et al. 2021), and the proxy used for convective storms in this study (CAPE) is also a proxy for spatial variation in lightning frequency. While the future of tropical lightning frequency remains debated (Romps 2019), reconstructions of historic lightning regimes in central Africa suggest that lightning frequency has been increasing for decades (Harel and Price 2020). If spatial associations between CAPE and large windthrow events are predictive of all windthrow (as implied in this study), then they should also be considered predictive of lightning frequency and lightning disturbance rates.

Minor comments

Lines 178-186. If convective storm frequency is increasing with climate change, then it likely has been increasing over the past decades as well. This is supported by the findings of Harel and Price (2020) in central Africa. It might be worth mentioning that an increase in convective storms may be contributing to the observed increases in tree mortality and weakening of the carbon sink across Amazonia.

Fig. S5 – This figure is excellent and provides important geographic information that will be of interest to anyone working in the Amazon. Pairing panel a from this figure with a second panel depicting a map of the Amazon with geographic variation based on the ensemble mean 'Delta-disturbance density' predictions from Table 1 (ideally from an improved model of the relationship between CAPE and disturbance density) would be more meaningful to the reader than the current Table 1. I suggest making this replacement.

References:

- Chambers, J. Q., R. I. Negron-Juarez, D. M. Marra, A. Di Vittorio, J. Tews, D. Roberts, G. H. Ribeiro, S. E. Trumbore, and N. Higuchi. 2013. The steady-state mosaic of disturbance and succession across an old-growth central amazon forest landscape. *Proceedings of the National Academy of Sciences* 110:3949-3954.
- Gora, E. M., J. C. Burchfield, H. C. Muller-Landau, P. M. Bitzer, and S. P. Yanoviak. 2020. Pantropical geography of lightning-caused disturbance and its implications for tropical forests. *Global Change Biology* 26:5017– 5026.
- Gora, E. M., P. M. Bitzer, J. C. Burchfield, C. Gutierrez, and S. P. Yanoviak. 2021. The contributions of lightning to biomass turnover, gap formation and plant mortality in a tropical forest. *Ecology* 102:e03541.
- Harel, M., and C. Price. 2020. Thunderstorm trends over africa. *Journal of Climate* 33:2741-2755.
- Romps, D. M., J. T. Seeley, D. Vollaro, and J. Molinari. 2014. Projected increase in lightning strikes in the United States due to global warming. *Science* 346:851-854.
- Romps, D. M. 2019. Evaluating the future of lightning in cloud-resolving models. *Geophysical Research Letters* 46:14863–14871.
- Yanoviak, S. P., E. M. Gora, P. M. Bitzer, J. C. Burchfield, H. C. Muller-Landau, M. Detto, S. Paton, and S. P. Hubbell. 2020. Lightning is a major cause of large tropical tree mortality in a lowland neotropical forest. *New Phytologist* 225:1936-1944.

Reviewer #2 (Remarks to the Author):

In the manuscript titled "Climate warming projected to cause a large increase in Amazon wind throw disturbance this century", the authors suggest a methodology to estimate future changes in wind throw events in the Amazon forest based on CAPE as a predictor for severity of convective storms, which CMIP models (different from the convective storms themselves) can represent. The manuscript touches upon a very important topic, linking the changes in convective storms to their potential effects on ecosystem structures. However, the manuscript is on the one hand lacking information in the methodology description, and on the other hand seems to oversimplify relationships between mean CAPE changes and convective storm changes (even without

going into the complexity of connecting CAPE to downdraft and wind gust strength) without acknowledging limitations. As the references in this paper detail, while we know that convective storms are important for windthrow, and the expected future upwards trend in CAPE across the tropics is known, the direct consequences for windthrow remain tricky to quantify. The novelty here hence lies in the attempt to translate change in large-scale CAPE into change in windthrow cases.

While I think the posed question is very important, I don't see that the quantification as done here can be justified based on used datasets. Specifically, the balance between intensity and frequency change in future storms is not discussed, which however would directly affect wind throw cases (imagine more extreme storms, but fewer storms overall). Based on increased mean CAPE, future windthrow events could just as well become less frequent but larger in area on a case basis (e.g. by larger, individually more intense storms).

This paper could be published, but in a more CAPE-conceptual rather than factual fashion with necessary uncertainty and limitation discussion (due to the unknowns regarding storm frequency changes and a lack of convection-permitting projection data to test storm process behaviour i.e. storm intensity and frequency scaling with CAPE for the region). The manuscript hence lacks quality in this regard. Better explanation of parts of the applied scaling strategy from CAPE to wind throw cases and a major overhaul of wording and explicit detailing of the taken assumptions (and limitations therein!) to arrive at conclusions presented here is necessary. At this point, the methods section also only offers mediocre quality descriptions. Finally, the point of the potential connection between storm intensification and forest structures is well worth making and I would be happy to see this question better constrained, but I don't think this study uses the necessary tools (nor adapted wording to account for the limited tools) to take a considerable step forward, producing hand-wavy results. I am hence not sure this study fulfills Nature Communication requirements.

Major comments:

- The study heavily relies on the assumption that the relationship between mean CAPE in a region and the frequency of storms of sufficient strength remains the same now and in a future climate. Unfortunately, this is a difficult question to answer without consulting a convection-permitting model projection for this regional domain. As the authors note, we are pretty sure that as CAPE extremes increase, storm extremes will increase in a warming climate. However, considering the full distribution, it is much harder to say how storm intensity and in fact storm occurrence will be linked to CAPE. The authors here assume that relationships between CAPE and wind throw cases (i.e. proxy for storm frequency) will remain the same. However, convection-permitting models tend to show in many regions that, while extreme storm frequency increases, storm days decrease. We may not be looking at a simple upward shift of CAPE distributions in the future, but at a flatter, broader curve (refer to Prein et al. 2017 for example, their Fig. 4e), i.e. decrease in number of medium events while intense events increase. And indeed, translating these distributions into storm intensities may show that even similar CAPE values in the future produce more intense storms (refer to Fitzpatrick et al. 2020, their Fig7f), which means that the "future minus current"-CAPE delta most likely doesn't follow the current wind throw regression with CAPE.

In any case, change in mean CAPE then does not reflect these possible relationship changes between CAPE and storm frequency.

The key sentence is hidden in methods section:

315-316 "it is assumed that the windthrow density as a function of CAPE does not change"

with windthrow density == storm frequency per area (by proxy).

How do the authors justify this assumption not supported by convection-permitting model results, an assumption that should form part of the discussion in the main text?

- It seems the dominant explanatory power by using mean CAPE in the non-parametric model is the fact that for the first bin (CAPE < ~ 1000) very few storms occur

(probability for windthrow = 0 without storm), while sensitivity of windthrow to CAPE afterwards flattens off. What is the contribution of zero CAPE days here? Would the authors say that the windthrow relationship to CAPE is dominated by the binary question "is there a storm, yes or no" rather than increasing storm intensity with CAPE? Given that global warming may rather decrease storm number but increase intensity when one occurs, it is important to understand whether frequency or intensity is dominating the model skill here. Have the authors checked how observed storm frequency or intensity relates to CAPE bins?

The question of variables affecting storm wind gust potential:

Please refer to Brown and Dowdy 2021 <https://doi.org/10.1029/2021JD034633>, "Severe Convective Wind Environments and Future Projected Changes in Australia" for variable evaluation more directly related to wind gust potential. I don't suggest to follow such complex analysis, but simplifications in this study here must be mentioned even if space is limited to help result interpretation.

Fig2d: remove the fitting line, negative values for <400 are not valid, or in fact I'd suggest to drop this model and focus on discussing the considerable area increase in >1000 J/kg mean CAPE regions, illustrating the more robust result that wind throw will potentially affect a much larger ((six-fold?) Amazon area at levels that we only see for a smaller part of the forest today.

Methods section problems (non-exhaustive, generally hard to extract necessary information):

- 11296-302" Since different approaches were used to calculate ERA 5 CAPE and CMIP 6 CAPE 27, these two datasets are not comparable. Therefore, we generated the delta CAPE map for each model by calculating the ratio between CMIP 6 future CAPE and CMIP 6 current CAPE, and the delta CAPE map indicated the projected increase in CAPE from 1990-2015 to the end of 21st century. Then we applied the delta CAPE to the ERA 5 mean afternoon CAPE by multiplication to scale the future CAPE and make it comparable with ERA 5 current mean afternoon CAPE.

>>what does it mean? I assume CAPE changes were calculated as the CMIP difference relative to current values: $[\text{future}[\text{cmip}] - \text{current}[\text{cmip}]] / \text{current}[\text{era5}] (\%)$?
Next, what was multiplied? What's the slope (unit)? What is the valid value range for applied "model" (which by the way seems to falls apart for CAPE values < 400 J /kg, when it predicts negative values for windthrow and is thus invalid for quite a few boxes in Fig1d - how is this considered in the calculations?).

Please provide equations for all your scaling calculations and the results presented in the table.

Table 1: Parametric model: how exactly are the change values in Table 1 calculated? Furthermore, what justifies the averaging of the two (often vastly different) "statistical / parametric" model results to conclude on "likely future windthrow cases?"

Fig2b: information on ensemble uncertainty missing

Fig.2d: just to be sure, could the authors check whether this is really r^2 and not r ? It is somewhat surprising that this should result in 57% explained variance.

Minor points:

- What is the reason to average CAPE for 13-1900LT, including periods when it is being consumed by active convection, rather than focusing on max. CAPE in the pre-convective period (say 10-14LT?)

- Please give the information what CAPE is considered (how it's calculated) for CMIP6 and in ERA5 respectively.

- Fig1: It would be useful to plot the windthrow events from (a) in some way onto (b) for easier pattern comparison. The patterns do not seem all that obvious. In addition, could the authors clarify why the western parts of Amazonia show remarkably few wind throw cases while the authors of reference (1) identify comparably high uprooted tree mortality per year there (e.g. their Fig2d) with respect to the other regions? Is this a sampling issue?
(1) Esquivel-Muelbert, A. et al. Tree mode of death and mortality risk factors across Amazon forests. Nat. Commun. 11, (2020).

II.176 Please note in text that provided uncertainties are based on 1 standard deviation

II 43-44: " CAPE determines the speed limit of upward convection." , which means it could also be less (entrainment), or in fact more if conditions are strongly sheared <https://doi.org/10.1175/JAS-D-20-0103.1> I don't want to get into the question of not considered shear here though. There are bigger problems in this manuscript. However, the sole focus on CAPE for storm intensity should at least be discussed and the exclusion of other possibly important variables mentioned.

REVIEWER COMMENTS

Reviewer #1 (Remarks to the Author):

This study quantifies the relationship between large windthrow events, as detected visually via Landsat images, and a proxy for convective storm activity (CAPE) across the Amazon basin. Aggregating data at the 2.5 degree scale revealed a strong relationship between CAPE and windthrow density across the Amazon basin for the last 30 years. This spatial relationship is then combined with future climate models to predict how windthrow events will change across the next century as a function of changes in CAPE. The estimates vary, but all show a substantial increase in windthrow frequency that has major consequences for future forests. The study is well written and succinctly addresses a major knowledge gap. Overall, the results are very important and have the potential to meaningfully improve how scientists think about the ways that changing climate affects tropical forests. In particular, this work is meaningful because it highlights an understudied aspect of global change that could play a dominant role in shaping future forests. However, I do have some major comments and concerns related to various parts of the manuscript.

Thank you so much for the recognition of our work. We really appreciate your comments and suggestions.

We have made major improvements based on your comments as follows:

1. added discussions on:
 - potential biases arising from disturbance identification and sampling error
 - biases in regional variation in vegetation composition and dynamics
2. had a better documentation of methods section with more details
 - clarified the windthrow identification methods
3. dropped the statistical model and focused on the spatial pattern of increases in the area favorable to extreme storms
4. stated clearly in introduction section that this paper focused on large windthrow (>25,000 m²) that produced by extreme convective storms

Major comments

1. There is substantial potential for bias based on the current description of the methods for identifying windthrows. Concerns related to these potential biases should be addressed, potentially through a combination of clarifications to the methods and supporting analyses. Based on the description provided in the methods (lines 256-268), it is not clear how the authors handled biases in (1) the spatial distribution of image selection across the region (i.e., were the 395 images used in this study evenly distributed across the Amazon? Or were some areas sampled more than others?);

Thanks for the question. 395 images used in this study are evenly distributed to cover the Amazon region. Please find the alignment of the images in Fig. S5 (see below). Landsat images are displayed in false color: R: SWIR (Band 6); G: NIR (Band 5); B: Red (Band 4). Hollow regions indicates that we cannot find cloud-free (<20%) image in that region in 2019.

Figure S5. Landsat scenes used to identify 1012 windthrows. Map showing Landsat images covering the Amazon used in this study.

We have revised the methods part to include all the detailed steps.

[Line 280-286, Methods] Landsat images from January 1st 2018 to December 31st 2019 were filtered on 20% or less of cloud coverage, and only the least cloudy image at each location was selected to make an image composite covering the entire Amazon region. In total, 395 least cloudy Landsat 8 images within the Amazon boundary during 2018-2019 were selected and displayed in false color (red: SWIR band, green: NIR band, blue: red band) on Google Earth Engine for windthrow events identification (Fig. S5). Hollow regions on Fig. S5 (2.8% of the total area of the Amazon region) indicated that no clear images with less than 20% cloud were found in these areas.

(2) temporal differences in image selection (for example, if two regions have 6 images each, were those images distributed evenly across 30 years in one region but clustered in the same 6-year period in the second region?);

As stated in the response to last question, we only used the least cloudy image chosen from 2018-2019 to make the composite that can cover the whole Amazon region.

(3) biases in visual disturbance identification and sampling effort among images (i.e., were all images visually scanned for the same amount of time? Were they scanned until new windthrow detection rates dropped below a certain level? Or standardized in some other way?);

Thanks for raising these questions. We have addressed the biases in disturbance identification and sampling effort in discussion.

[Line 233-243, Discussion] This first step in projecting the impact of future extreme wind events on Amazon forests has several caveats, which we discuss here to facilitate the interpretation of our results. Potential biases can exist in disturbance identification and sampling effort. 2.8% of the Amazon region was not included in this study due to the lack of clear remote sensing images (clouds < 20%). Windthrows without a characteristic shape were eliminated from this study. We

only considered the spatial pattern of large windthrows in this study while omitting vast majority of windthrows which are smaller than 25,000 m². Since the focus of this study is on the spatial pattern of large windthrow events, the underestimation of windthrow events will not have significant impact on the results. In addition, the large windthrow dataset in this study provides a comprehensive view over the entire Amazon which would be difficult to achieve using small-scale windthrow datasets.

We used spectral mixture analysis as the standard way to identify windthrows. We have added more details in methods part.

[Line 286-297, Methods] Spectral mixture analysis was applied on each image to detect potential windthrows (similar methods have been used in previous windthrow studies^{5,17}). Each potential windthrow was then visually checked and evaluated based on authors' 15-year experience working with windthrow and remote sensing^{2,3,17,19}. "New" windthrows that occurred within 1 year were spectrally more visible based on their clear fan-shape^{5,10} (diverging from a central area with small pixels scattered at the tail) and their relatively distinguishable reddish colors (Fig. S6a), while "old" windthrows (Fig. S6b) occurred over 1 year from the image of identification, that have bright green colors, were also verified using historical Landsat images that can go as far as 1984 (when Landsat 5 was launched). 10-15% of "old" windthrows without fan-shape were eliminated from this study because it was hard to identify if they were windthrows or other types of forest disturbance. The minimum size of windthrows that identified in this study was 25,000 m².

"New" windthrow:
centered at -74.25634° , -3.75625° ,
displayed in Landsat composite,
displayed in false color (red: SWIR
band, green: NIR band, blue: red
band) from 2018-2019, the
windthrow occurred in 07/30/2017

"Old" windthrow:
centered at -59.12147° , -2.81343° , displayed
in Landsat composite, displayed in false
color (red: SWIR band, green: NIR band,
blue: red band) from 2018-2019, the
windthrow occurred in 10/16/2000

Fig. S6 (a) "New" Windthrow that occurred within 2 years of identification. The reddish color indicated that the windthrow was relatively fresh with lots of dead trees with high reflectance in SWIR; (b) "Old" windthrow that occurred nearly 20 years ago from identification in satellite images.

or (4) biases arising from regional variation in vegetation composition or vegetation dynamics (e.g., high productivity obscuring disturbances more rapidly).

We thank the reviewer for this observation. As we shown in our previous studies, windthrows in the western Amazon recover faster than in the Central Amazon. Our study does not address the dynamics of windthrows (a topics we are developing in another study in progress) but their occurrence. This occurrence revealed clear spatial patterns that we correlated with spatial patterns of CAPE.

We have added the discussion on biases arising from regional variation in vegetation composition or vegetation dynamics.

[Line 245-256, Discussion] There are also biases arising from regional variation in vegetation composition or vegetation dynamics. Compared with central Amazon, forests in northwestern Amazon were more vulnerable to windthrow mortality and recover faster due to a number of factors, including tree stem density, soil type, root architecture, tree diversity, microclimate condition, topographic exposure^{2,39}. A previous study showed that windthrows were spatially more common in northwestern Amazon than in central Amazon². With high windthrow tree mortality, forest in northwestern amazon took about 20 years to recover to 90% of “pre-disturbance” biomass, and forests in central Amazon took about 40 years to recover, and the biomass recovery depends on the windthrow severity and time since disturbance^{32,39}. Therefore, fewer large windthrows may be identified in this study in northwestern Amazon due to relatively faster recovery. More accurate windthrow identification could be done at a yearly basis. Such analysis, however, is beyond the scope of the current study.

All four of these factors could influence spatial variation in the disturbance density metrics used in this study. Explicitly state and, when possible, quantify the potential biases associated with each of these three issues. If biases are identified, it will be important to estimate and correct for these biases.

Thanks for the comments on four aspects of biases. They are very helpful in making our results more robust. We have addressed all the questions in methods and discussion sections.

2. The linear regression expressed in Fig. 2d is not appropriate for the relationship between windthrow density and CAPE. Indeed, the current relationship predicts impossible negative values for all CAPE under 400 J kg⁻¹. Windthrow density cannot possibly be negative, so the fitted relationship should approach an asymptote at 0 on the y-axis (or the intercept should be fixed at 0, but that doesn't seem as though it would produce a good fit). Please provide a more realistic form for the relationship between these two variables. This is particularly important because the strength of this linear relationship and its future projections are the two primary results of this study.

Thanks for the comments. The second reviewer also had the similar comments. The second reviewer suggested dropping the linear regression model and focusing on discussing the increase in area with storm-favorable condition. Therefore, we deleted the linear model and focused on the increase in the area favorable to extreme storms (CAPE > 1023 J kg⁻¹), which indicates a

more robust result that convective storms will affect a much larger area ($51\% \pm 20\%$) of forests in Amazon by the end of this century.

We have updated Fig.2c and revised the results section based on both of your suggestions.

[Line 158-161, Results] 38% of Amazon region has a mean CAPE over 1023 J kg^{-1} (Fig. 2c), which provides a storm-favorable condition with high windthrow density. In general, windthrows occur nearly 8 times more frequently in the favorable environments (CAPE $>1023 \text{ J kg}^{-1}$, hereafter storm-favorable environments) than in the non-favorable environments (CAPE $\leq 1023 \text{ J kg}^{-1}$).

[Line 181-191, Results] While the fraction of area that is storm-favorable in the current mean CAPE map is 38%, ESMs project that $58 \pm 8\%$ of the Amazon region becomes storm-favorable by the end of the 21st century. The area with CAPE over 1023 J kg^{-1} increases $51\% \pm 20\%$ by the end of the the 21st century (Table 1), indicating a much larger area with storm-favorable conditions within the Amazon. An area of nearly $1,390,000 \text{ km}^2$ in southern and central Amazon, including forested areas in the northern part of Bolivia, the south-eastern part of Peru, and central Brazil, where windthrows are infrequent now, are projected to experience a sixfold or greater increase in windthrow events by 2100 (Fig. 2c). North-western and western Amazon, where large windthrow density are relatively frequent now (2.4 windthrows/ $10,000 \text{ km}^2$), are estimated to experience 33%-50% increase in the number of windthrow events in the next 50-80 years (Fig. S4c).

Fig. 2 c) The increase in area with CAPE over 1023 J kg^{-1} , with orange pixels representing mean 1990-2019 ERA 5 CAPE higher than 1023 J kg^{-1} and red pixels representing mean scaled 2070-2099 CMIP6 CAPE higher than 1023 J kg^{-1}

3. Be more transparent about what is being measured in this study and its implications. The manuscript consistently implies that this study measures the relationship between convective storms and total windthrow rates by omitting information about the windthrow events being measured. Specifically, it is not explicitly stated that all relationships focus on uncommonly large windthrow events that make minor contributions to total tree mortality and total windthrow

rates, as shown by the authors' previous work (Chambers et al. 2013). It is likely that the authors chose to focus only on large windthrow events because this is feasible, and that decision is reasonable. However, it should be explicitly stated in the introductory text (potentially in the paragraph on lines 72-81) that this study uses large windthrow events as a tractable proxy for all windthrow. Similarly, the abstract and results should explicitly state that his study shows relationships between large windthrow events and CAPE, not 'windthrow' without qualification. Somewhere early in the text it should be stated that this approach omits the vast majority of windthrow events (Fig. 2 of Chambers et al. 2013).

Thank you so much for the suggestions. We have explicitly stated "large windthrows" and explained it using your words in the introduction section.

[line 80-83, Introduction]: This study seeks to investigate the spatial pattern of windthrow events that are larger than 25,000 m² (hereafter 'large windthrows') and their correlation with CAPE over 30 years while demonstrating that CAPE is an appropriate proxy for estimating the spatial pattern of large windthrow events in the Amazon.

[Line 95-99, Introduction] This study uses only large windthrow events that can be identified using Landsat satellite images. These large windthrows are produced by extreme convective storms¹⁰, such as squall lines¹⁷. Although this approach omits the vast majority of small windthrow events ($\leq 25,000 \text{ m}^2$)¹⁹, the general spatial pattern of large windthrows occurrence is used as a tractable proxy for all windthrows.

Lastly, based on the authors' previous work, it should be possible to approximate the proportion of total windthrow events (or the proportion of total disturbed area) represented by the windthrow events included in this study. Even if it was imprecise, such an estimate would give the reader clarity about the detection efficiency of windthrow events used in this study.

Thanks for the suggestion. We have added the estimate in the methods section.

[Line 299-302, Methods] Based on a gap-size probability distribution function (PDF) that simulates the entire disturbance gradient from all sizes of windthrows¹⁹, the proportion of total tree mortality represented by large windthrows ($>25,000 \text{ m}^2$) identified in this study is 0.5-1.1%.

4. Similar to point 3, some statements about the importance and novelty of this work are a little misleading. This study is meaningful on its own, and these overstatements unnecessarily undermine the important findings. I highlight two examples here, but it would be good to be more cautious throughout the text.

First, the opening sentence states that windthrows cause more than 50% of tree mortality in Amazonia. The study being cited (Esquivel-Muelbert et al. 2020) estimates breakages and uprooting as 51% of mortality across Amazonian forest plots with a 95% confidence interval of 48-54%. I doubt anyone reading 'more than 50%' interprets this to mean that the chances of being at or below 50% are as nearly as high as above 50%. This overstatement is unnecessary. Second, the abstract states that, "This relationship builds connections between strong convective storms and forest dynamics for the first time in the Amazon." This is not true. Previous work

showed that lightning frequency (also associated with convective storms, as demonstrated by decades of research on thunderstorms and by Dr. Romps previous work on CAPE and lightning; Romps et al. 2014) was strongly associated with forest dynamics across the Amazon, in terms of fewer large trees, high rates of woody biomass loss, and less aboveground biomass in forests with higher lightning frequency (Gora et al. 2020). Narrowing the focus on the patterns analyzed in this study would avoid overstating the findings.

Thanks for your comments. We have fixed the texts based on your suggestion. We have also narrowed our focus on the spatial pattern.

[Line 17-19, Abstract] Here we develop empirical relationships and a conceptual model that maps convective available potential energy (CAPE), which is well simulated by ESMs, to the spatial pattern of large windthrow events.

[Line 19-20, Abstract] This relationship builds connections between strong convective storms and forest dynamics in the Amazon.

[Line 28, Introduction] Convective storms account for around 50% of the tree mortality across Amazonia ...

5. The methods for image selection (lines 258-268) were not detailed enough to facilitate reproducibility. There were a number of issues, listed with letters below:

a. Since 30x30m is the size of a Landsat pixel, the description of windthrow selection suggests that any event with at least one pixel could be included. However, that doesn't seem to be the case based on the low number of windthrow events. Clarify the minimum size threshold used here.

We have added the details on the minimum size (25000 m²) of windthrows in this study as addressed in questions above.

b. Referring to Fig. S6 for clarification, it is not clear whether the disturbances shown in each image were considered one windthrow event or how this approach would differentiate nearby windthrow events from each other. In general, there is a need for more detail about how windthrow events were classified and how they were separated from each other and from other disturbance events.

We have added more explanation in Fig.S7 (Fig. S6 in original version) caption.

[Line 53-57, revised supplementary material] Fig. S7. Three windthrow events displayed in the pink color on Landsat 5 TOA images with false color visualization (red: SWIR, green: NIR, blue: RED). Each windthrow displayed shows a clear fan shape diverging from a head with scattered small disturbance pixels spreading at the tail, indicating wind direction from windthrow head to tail. This distinctive shape helps authors determine the windthrow event and separate it from other type of disturbances.

c. As for assigning the date of occurrence, it is not clear how the date was determined from the text or why the date of occurrence for 125 disturbances indicated that all 1012 disturbances occurred between 1990-2019. If these 125 dated disturbances were used to identify similarly aged disturbances, then provide details about how many disturbances were removed because of an age mismatch. If none were removed, explain why this method should be viewed as effective given the probability that a random sample of images would include disturbances from a prior period.

Thanks for the question. We didn't explain very clearly in the original version and we have rewritten the methods section to explain this question.

[line 307-313, Methods]: Previous studies showed that windthrows in northwestern Amazon took about 20 years to recover to 90% of "pre-disturbance" biomass from all damage classes while forests in central Amazon took about 40 years to recover³⁶. The biomass recovery depends on the windthrow severity and time since disturbance³⁷. Therefore, we estimated that these 1012 windthrows most likely occurred within 30 years (between 1990-2019), and the estimated occurrence period was validated using the occurrence year of 125 windthrow cases.

d. There may be a word missing from the sentence on lines 258-261 (potentially an "and" before "an area over"?), and as a result, it is not clear if all 1012 windthrow events have all three characteristics described in this sentence.

Thanks a lot for reading so carefully. The word "and" was missing and we added it.

6. The study should acknowledge the importance of lightning as an agent of forest disturbance associated with convective storms. Lightning is mentioned once, but its role as an agent of disturbance is ignored. Lightning could play a key role in the effects of changing convective storm frequency on future forests and acknowledging that consequence would strengthen this study. Specifically, lightning was recently shown to be a major agent of canopy disturbance and large tree mortality (Yanoviak et al. 2020; Gora et al. 2021), and the proxy used for convective storms in this study (CAPE) is also a proxy for spatial variation in lightning frequency. While the future of tropical lightning frequency remains debated (Roms 2019), reconstructions of historic lightning regimes in central Africa suggest that lightning frequency has been increasing for decades (Harel and Price 2020). If spatial associations between CAPE and large windthrow events are predictive of all windthrow (as implied in this study), then they should also be considered predictive of lightning frequency and lightning disturbance rates.

Thanks for the suggestion. We have acknowledged the importance of lightning in creating forest disturbance in Introduction section.

[Line 63- 65, Introduction] In addition to wind and rainfall, lightning associated with convective storms can also be a major agent of canopy disturbance and large tree mortality^{21,22}.

We would like to make it clear that although lightning is an agent of forest disturbance associated with extreme convective storms, it is quite different from windthrow disturbance.

First, lightning only results in standing dead trees (Yanoviak et al., 2020) while most of the tree in windthrown area were uprooted or snapped (Nelson et al., 1994). Second, lightning results in a much smaller scale of tree mortality (each strike kill 3.5 trees directly with 11.4 indirectly in Yanoviak et al., 2020) while large windthrows result in cluster of downed tree (forest gaps at a minimum of 2,5,000 m², which indicates few hundreds dead trees in using binned PDFs in Chambers et al., 2013). Third, according to our previous field observations, we did not see lightning killed trees in windthrown areas. Therefore, further analysis on lightning and forest disturbance are not included to avoid any confusion on windthrow and lightning disturbances.

Minor comments

Lines 178-186. If convective storm frequency is increasing with climate change, then it likely has been increasing over the past decades as well. This is supported by the findings of Harel and Price (2020) in central Africa. It might be worth mentioning that an increase in convective storms may be contributing to the observed increases in tree mortality and weakening of the carbon sink across Amazonia.

Thanks for the suggestions. We have added it to the discussion.

[Line 216-218, Discussion] The relationship between extreme storms and tree mortality also imply that increasing convective storms³² may contribute to the observed increase in tree mortality and a weakening of the carbon sink across Amazonia³³⁻³⁵.

Fig. S5 – This figure is excellent and provides important geographic information that will be of interest to anyone working in the Amazon. Pairing panel a from this figure with a second panel depicting a map of the Amazon with geographic variation based on the ensemble mean ‘Delta-disturbance density’ predictions from Table 1 (ideally from an improved model of the relationship between CAPE and disturbance density) would be more meaningful to the reader than the current Table 1. I suggest making this replacement.

Thanks. We have followed your suggestions and replaced fig.2d with S5 (the spatial pattern of increase in future CAPE). We have also created a new figure (Fig. 2c) comparing the increase in area of storm-favorable area (CAPE > 1023 J kg⁻¹) in Amazon.

Fig. 2 d) Ensemble-mean increase of estimated CAPE from the current climate (1990-2014) to the future climate (2070-2099) under the SSP585 scenario. Since CMIP 6 models provide historic simulations only up to 2015, data from 2015-2020 are not included. Stippling indicates

regions where all 10 ESMs agree on the increase of CAPE, with CAPE calculated using daily surface pressure and atmospheric profiles at standard pressure levels;

References

- Yanoviak, S. P., Gora, E. M., Bitzer, P. M., Burchfield, J. C., Muller-Landau, H. C., Detto, M., Paton, S., & Hubbell, S. P. (2020). Lightning is a major cause of large tree mortality in a lowland neotropical forest. *New Phytologist*, 225(5), 1936–1944. <https://doi.org/10.1111/nph.16260>
- Nelson, B. W., Kapos, V., Adams, J. B., Oliveira, W. J., Braun, O. P. G., & do Amaral, I. L. (1994). Forest disturbance by large blowdowns in the Brazilian Amazon. *Ecology*, 75(3), 853–858. <https://doi.org/10.2307/1941742>
- Chambers, J. Q., Negron-Juarez, R. I., Marra, D. M., Di Vittorio, A., Tews, J., Roberts, D., Ribeiro, G. H. P. M., Trumbore, S. E., & Higuchi, N. (2013). The steady-state mosaic of disturbance and succession across an old-growth central Amazon forest landscape. *Proceedings of the National Academy of Sciences of the United States of America*, 110(10), 3949–3954. <https://doi.org/10.1073/pnas.1202894110>

Reviewer #2 (Remarks to the Author):

In the manuscript titled “Climate warming projected to cause a large increase in Amazon wind throw disturbance this century”, the authors suggest a methodology to estimate future changes in wind throw events in the Amazon forest based on CAPE as a predictor for severity of convective storms, which CMIP models (different from the convective storms themselves) can represent. The manuscript touches upon a very important topic, linking the changes in convective storms to their potential effects on ecosystem structures. However, the manuscript is on the one hand lacking information in the methodology description, and on the other hand seems to oversimplify relationships between mean CAPE changes and convective storm changes (even without going into the complexity of connecting CAPE to downdraft and wind gust strength) without acknowledging limitations. As the references in this paper detail, while we know that convective storms are important for windthrow, and the expected future upwards trend in CAPE across the tropics is known, the direct consequences for windthrow remain tricky to quantify. The novelty here hence lies in the attempt to translate change in large-scale CAPE into change in windthrow cases.

While I think the posed question is very important, I don't see that the quantification as done here can be justified based on used datasets. Specifically, the balance between intensity and frequency change in future storms is not discussed, which however would directly affect wind throw cases (imagine more extreme storms, but fewer storms overall). Based on increased mean CAPE, future windthrow events could just as well become less frequent but larger in area on a case basis (e.g. by larger, individually more intense storms).

This paper could be published, but in a more CAPE-conceptual rather than factual fashion with necessary uncertainty and limitation discussion (due to the unknowns regarding storm frequency changes and a lack of convection-permitting projection data to test storm process behaviour i.e. storm intensity and frequency scaling with CAPE for the region). The manuscript hence lacks quality in this regard. Better explanation of parts of the applied scaling strategy from CAPE to wind throw cases and a major overhaul of wording and explicit detailing of the taken assumptions (and limitations therein!) to arrive at conclusions presented here is necessary. At this point, the methods section also only offers mediocre quality descriptions. Finally, the point of the potential connection between storm intensification and forest structures is well worth making and I would be happy to see this question better constrained, but I don't think this study uses the necessary tools (nor adapted wording to account for the limited tools) to take a considerable step forward, producing hand-wavy results. I am hence not sure this study fulfills Nature Communication requirements.

We appreciate your feedback. We have made major improvements based on your suggestions as follows:

1. added discussions on:
 - biases in regional variation in vegetation composition and dynamics
 - model assumption and uncertainty due to the lack of convection-permitting models
 - limitations in storm intensity and forest structure changes
2. had a better documentation of methods section with more details

- improved the meteorological data calculation methods with necessary equations and calculations
3. dropped the statistical model and focused on the spatial pattern of increases in the area favorable to extreme storms
 4. switched the language in the texts to referring to the model as conceptual

Major comments:

- The study heavily relies on the assumption that the relationship between mean CAPE in a region and the frequency of storms of sufficient strength remains the same now and in a future climate. Unfortunately, this is a difficult question to answer without consulting a convection-permitting model projection for this regional domain. As the authors note, we are pretty sure that as CAPE extremes increase, storm extremes will increase in a warming climate. However, considering the full distribution, it is much harder to say how storm intensity and in fact storm occurrence will be linked to CAPE. The authors here assume that relationships between CAPE and wind throw cases (i.e. proxy for storm frequency) will remain the same. However, convection-permitting models tend to show in many regions that, while extreme storm frequency increases, storm days decrease. We may not be looking at a simple upward shift of CAPE distributions in the future, but at a flatter, broader curve (refer to Prein et al. 2017 for example, their Fig. 4e), i.e. decrease in number of medium events while intense events increase. And indeed, translating these distributions into storm intensities may show that even similar CAPE values in the future produce more intense storms (refer to Fitzpatrick et al. 2020, their Fig7f), which means that the “future minus current”-CAPE delta most likely doesn’t follow the current wind throw regression with CAPE.

In any case, change in mean CAPE then does not reflect these possible relationship changes between CAPE and storm frequency.

The key sentence is hidden in methods section:

315-316 “it is assumed that the windthrow density as a function of CAPE does not change” with windthrow density == storm frequency per area (by proxy).

How do the authors justify this assumption not supported by convection-permitting model results, an assumption that should form part of the discussion in the main text?

Thanks for your comments. In response to your comments, we have reworded the language from “the increase in future windthrow density” to “the increase in storm-favorable area ($CAPE > 1023 \text{ J kg}^{-1}$)”. We have switched the language in the texts referring the model as conceptual rather than factual based. As suggested by the first reviewer, we have focused more on analyzing the spatial pattern of future CAPE in this study and added results on the spatial pattern of increased storm-favorable area.

We acknowledge that the frequency of extreme convective weather will increase while weak and moderate convective storms will decrease (Rasmussen et al., 2020). We would like to explain that 1012 large windthrows ($>25,000 \text{ m}^2$) that we discussed in this paper can only be produced by extreme convective storms (Garstang et al., 1998), such as squall lines (Garstang et al., 1994; Negrón-Juárez et al., 2010). Therefore, the results produced by convection-permitting models show the increase in the frequency of extreme storms (with reflectivity range 40–50, 50–60, 60–70 dBZ) (Rasmussen et al., 2020) could be a supportive evidence of the increase in future storm-favorable condition and increase in future large windthrows in this study. We acknowledge that

the assumption may not apply to weak and moderate storms. We have added explanations on large windthrows and extreme windstorms in introduction. We have also addressed the limitations in the assumption, and uncertainty due to the lack of convection-permitting models in the discussion.

[Line 88-89, Introduction] A conceptual model is also developed using CAPE to project future windthrow events.

[Line 158-162, Results] 38% of Amazon region has a mean CAPE over 1023 J kg^{-1} (Fig. 2c), which provides a storm-favorable condition with high windthrow density. In general, windthrows occur nearly 8 times more frequently in the favorable environments (CAPE $>1023 \text{ J kg}^{-1}$, hereafter storm-favorable environments) than in the non-favorable environments (CAPE $\leq 1023 \text{ J kg}^{-1}$).

[Line 181-191, Results] While the fraction of area that is storm-favorable in the current mean CAPE map is 38%, ESMs project that $58 \pm 8\%$ of the Amazon region becomes storm-favorable by the end of the 21st century. The area with CAPE over 1023 J kg^{-1} increases $51\% \pm 20\%$ by the end of the the 21st century (Table 1), indicating a much larger area with storm-favorable conditions within the Amazon. An area of nearly $1,390,000 \text{ km}^2$ in southern and central Amazon, including forested areas in the northern part of Bolivia, the eastern part of Peru, and Brazil, where windthrows are infrequent now, are projected to experience a sixfold or greater increase in windthrow events by 2100 (Fig. 2c). North-western and western Amazon, where large windthrow density are relatively frequent now (2.4 windthrows/ $10,000 \text{ km}^2$), are estimated to experience 33%-50% increase in the number of windthrow events in the next 50-80 years (Fig. S4c).

[Line 80-83, Introduction] This study seeks to investigate the spatial pattern of windthrow events that are larger than $25,000 \text{ m}^2$ (hereafter ‘large windthrows’) and their correlation with CAPE over 30 years while demonstrating that CAPE is an appropriate proxy for estimating the spatial pattern of large windthrow events in the Amazon.

[Line 96-97, Introduction] This study uses only large windthrow events that can be identified using Landsat satellite images. These large windthrows are produced by extreme convective storms¹⁰, such as squall lines¹⁷.

[Line 258-267, Discussion] The function of CAPE and large windthrow density developed in this study is based on the assumption that increases in CAPE will result in increases in the frequency of extreme convective storms, which create more large windthrows. This assumption is supported by convection-permitting models that project an increase in extreme storm frequency with a warming climate⁴⁰. However, the assumption may not be applied to weak or moderate convective storms, which are projected to decrease in the future⁴⁰. Studies on small windthrows ($<25,000 \text{ m}^2$) and the corresponding convective storms can be carried out to explore their relationships. Future studies could involve convection-permitting models⁴¹ that are able to simulate MSCs to reduce the uncertainty in storm process behavior when exploring changes in storm frequency and intensity in the Amazon region.

- It seems the dominant explanatory power by using mean CAPE in the non-parametric model is the fact that for the first bin ($CAPE < \sim 1000$) very few storms occur (probability for windthrow = 0 without storm), while sensitivity of windthrow to CAPE afterwards flattens off. What is the contribution of zero CAPE days here?

The analysis we did does not shed any light on zero CAPE days. The CAPE values we are using are averaged over all days. We chose to expand the CAPE value in the step function to a minimum of 0 and a maximum of infinity. More details can be found in methods section.

[Line 377-380, Methods] The minimum and maximum of current ERA 5 mean afternoon CAPE was 42 and 1549. The minimum CAPE value of the first bin was extended to 0 and the maximum CAPE value of the last bin extended to infinity under the assumption that the windthrow density is similar for neighboring values.

Would the authors say that the windthrow relationship to CAPE is dominated by the binary question “is there a storm, yes or no” rather than increasing storm intensity with CAPE?

Yes, we agree with you and have made it clearer in the results section.

[Line 160-162, Results] In general, windthrows occur nearly 8 times more frequently in the favorable environments ($CAPE > 1023 \text{ J kg}^{-1}$, hereafter storm-favorable environments) than in the non-favorable environments ($CAPE \leq 1023 \text{ J kg}^{-1}$).

[Line 163-165, Results] The models shows that CAPE is a strong predictor of windthrow events spatial pattern in the Amazon region and provides a step function that can map CAPE values to large windthrow density.

Given that global warming may rather decrease storm number but increase intensity when one occurs, it is important to understand whether frequency or intensity is dominating the model skill here. Have the authors checked how observed storm frequency or intensity relates to CAPE bins?

Thanks for the question. We have focused more on the spatial pattern of increased storm-favorable area and address this question in the methods section.

[Line 380-382, Methods section] Since the windthrow density is roughly a step function in CAPE (see Fig. 2a), it is the increase in the area with high CAPE that then leads to an increase in the number of windthrow events.

The question of variables affecting storm wind gust potential:

Please refer to Brown and Dowdy 2021 <https://doi.org/10.1029/2021JD034633>, “Severe Convective Wind Environments and Future Projected Changes in Australia” for variable evaluation more directly related to wind gust potential I don’t suggest to follow such complex analysis, but simplifications in this study here must be mentioned even if space is limited to help result interpretation.

Thanks for the suggestions. We have added a paragraph in the discussion and emphasized the need to explore increasing CAPE with wind gust potential and precipitation rates. In addition, we are currently working on another manuscript of storm intensity and windthrow size.

[Line 269-275, Discussion] The study refrains from discussing the relationship between storm intensification and forest structure changes. How the increasing CAPE in Amazon region affects precipitation rate⁴² and wind gust potential⁴³, which are the major causes of tree mortality^{10,20}, has not been well explored. We emphasize the need for additional field observation datasets of severe convective winds and extreme rainfall rate associated with convective storms in the Amazon region. Such observations would improve the understanding of how forest dynamics link with convective storm intensity.

Fig2d: remove the fitting line, negative values for <400 are not valid, or in fact I'd suggest to drop this model and focus on discussing the considerable area increase in >1000 J/kg mean CAPE regions, illustrating the more robust result that wind throw will potentially affect a much larger ((six-fold?) Amazon area at levels that we only see for a smaller part of the forest today.

Thank you for the suggestion. This is really helpful for the model improvements. We have followed your suggestions and removed the statistical model. We have focused on discussing the considerable area increase in CAPE > 1023 J kg⁻¹ (shown in Fig 2c). We have also rewritten the results [Line 253-293, mentioned in answers to Question 1], illustrating a more robust result that storm-favorable area will increase 51% ± 20% in Amazon.

Fig 2 c) The increase in area with CAPE over 1023 J kg⁻¹, with orange pixels representing mean 1990-2019 ERA 5 CAPE higher than 1023 J kg⁻¹ and red pixels representing mean scaled 2070-2099 CMIP6 CAPE higher than 1023 J kg⁻¹;

Methods section problems (non-exhaustive, generally hard to extract necessary information):
- 11296-302“ Since different approaches were used to calculate ERA 5 CAPE and CMIP 6 CAPE 27, these two datasets are not comparable. Therefore, we generated the delta CAPE map for each model by calculating the ratio between CMIP 6 future CAPE and CMIP 6 current CAPE, and the delta CAPE map indicated the projected increase in CAPE from 1990-2015 to the end of 21st

century. Then we applied the delta CAPE to the ERA 5 mean afternoon CAPE by multiplication to scale the future CAPE and make it comparable with ERA 5 current mean afternoon CAPE. >>what does it mean? I assume CAPE changes were calculated as the CMIP difference relative to current values: $[\text{future}[\text{cmip}]-\text{current}[\text{cmip}]]/\text{current}[\text{era5}](\%)$? Next, what was multiplied? What's the slope (unit)? What is the valid value range for applied "model" (which by the way seems to falls apart for CAPE values < 400 J/kg, when it predicts negative values for windthrow and is thus invalid for quite a few boxes in Fig1d - how is this considered in the calculations?).

Thanks for the questions. We have rewritten the methods part and added detailed information.

[Line 339-346, Methods] We extracted daily surface temperature (tas), specific humidity (huss), surface pressure (ps), temperature (ta) from these models to calculate daily nondilute, near-surface-based, adiabatic CAPE. CMIP 6 CAPE was calculated by considering the buoyancy of a near-surface parcel lifted adiabatically to a series of discrete pressure levels (100 kPa to 10 kPa in increments of 10 kPa). CMIP 6 CAPE is calculated as follows:

$$CAPE = \sum_{i=1}^{10} dp H(b_i) b_i$$

Where $dp = 10$ kPa, H is the Heaviside unit step function, and $b_i = \frac{1}{\rho_i} - \frac{1}{\rho_{e,i}}$, with ρ_i being the parcel density at pressure level i and $\rho_{e,i}$ being the environmental density at pressure level i .

[Line 349-365, Methods] Since different approaches were used to calculate ERA 5 CAPE and CMIP 6 CAPE⁴⁵, the absolute CAPE values of the two datasets are not comparable. Therefore, for each ESM model, we scaled future CMIP 6 CAPE by multiplying, grid-wise, the delta CAPE generated from an individual model in CMIP 6 with the ERA 5 current mean afternoon CAPE (Fig. 1c) as follows:

$$\text{Delta CAPE} = (\text{CAPE}_{\text{CMIP6_future}} - \text{CAPE}_{\text{CMIP6_current}}) / \text{CAPE}_{\text{CMIP6_current}}$$

$$\text{CAPE}_{\text{scaled_CMIP6_future}} = \text{Delta CAPE} \times \text{CAPE}_{\text{ERA 5_current}}$$

The delta CAPE indicated the projected increase in CAPE from 1990-2015 to the end of the 21st century. In this way, a scaled CMIP 6 future CAPE map was generated for each model, and an ensemble-mean scaled CMIP6 CAPE map over 10 ESM models can be found in Fig. S4b. The scaled CMIP6 future CAPE values were comparable with ERA 5 current mean afternoon CAPE values, and both current and future CAPE maps were used to produce the increase in area with high CAPE values ($>1023 \text{ J kg}^{-1}$) in Table 1. The increase in area with storm-favorable environments was calculated as follows:

$$\text{Increase} = (\text{area_future} - \text{area_current}) / \text{area_current}$$

Where area_current is the area of $\text{CAPE} > 1023 \text{ J kg}^{-1}$ for current ERA 5 CAPE, and area_future is the area of $\text{CAPE} > 1023 \text{ J kg}^{-1}$ for the scaled CMIP6 future CAPE.

Please provide equations for all your scaling calculations and the results presented in the table.

Thanks for the reminder. We have provided the scaling calculations in the methods section mentioned in answers to previous question.

Table 1: Parametric model: how exactly are the change values in Table 1 calculated? Furthermore, what justifies the averaging of the two (often vastly different) “statistical / parametric” model results to conclude on “likely future windthrow cases”?

We have removed the statistical/parametric model as you suggested. Therefore, our main results focus on the increasing area of high CAPE that provides a favorable environment for more frequent extreme convective storms.

Fig2b: information on ensemble uncertainty missing

Thanks. We have added one standard deviation calculated from future CAPE of 10 ESMs used in this study.

Fig. 2 b) The area of the Amazon region in each CAPE bin for the past 30 years and for the last 30 years of the century. The error bars (SD) of future CAPE were generated using scaled 2070-2099 CMIP6 CAPE from 10 ESMs

Fig.2d: just to be sure, could the authors check whether this is really r^2 and not r ? It is somewhat surprising that this should result in 57% explained variance.

We have deleted the statistical model as you suggested.

Minor points:

- What is the reason to average CAPE for 13-1900LT, including periods when it is being consumed by active convection, rather than focusing on max. CAPE in the pre-convective period

(say 10-14LT?)

The main reason is that this study tries to focus on the general spatial pattern of CAPE and windthrows. 1012 windthrow cases were from 30 years (1990-2019), we think the average CAPE is an appropriate representation of general pattern over 30 years. Max CAPE is a good idea, but it can include extreme values and it is hard to define a certain period as pre-convective periods for all 1012 windthrows over 30 years.

- Please give the information what CAPE is considered (how it's calculated) for CMIP6 and in ERA5 respectively.

We have added more details on CMIP 6 and ERA 5 CAPE calculation in methods section.

[Line 327-329, Methods] ERA 5 CAPE was computed by considering parcels of air departing at different levels below the 35 kPa level, with maximum-unstable algorithm under a pseudo-adiabatic assumption⁴⁴.

[Line 341-342, Methods] CMIP 6 CAPE was calculated by considering the buoyancy of a near-surface parcel lifted adiabatically to a series of discrete pressure levels (100 kPa to 10 kPa in increments of 10 kPa).

More details on how to calculate CMIP 6 CAPE have been documented in [Line 343-346, Methods], which are addressed in the answers to previous questions.

- Fig1: It would be useful to plot the windthrow events from (a) in some way onto (b) for easier pattern comparison. The patterns do not seem all that obvious.

Thanks. We have edited figure 1c and plotted the contour lines of windthrow density on top of the CAPE to make it easier to identify the similar pattern.

Fig 1 (c) Contour lines of windthrow density (counts per 10,000 km²) over the mean afternoon CAPE at 0.25° resolution;

In addition, could the authors clarify why the western parts of Amazonia show remarkably few wind throw cases while the authors of reference (1) identify comparably high uprooted tree mortality per year there (e.g. their Fig2d) with respect to the other regions? Is this a sampling issue?

(1) Esquivel-Muelbert, A. et al. Tree mode of death and mortality risk factors across Amazon forests. Nat. Commun. 11, (2020).

Thanks for raising the concern on sampling biases. We have discussed the potential biases arising from sampling errors and from regional variation in vegetation composition or vegetation dynamics in discussion.

[Line 246-254, Discussion] Compared with central Amazon, forests in northwestern Amazon were more vulnerable to windthrow mortality and recover faster due to a number of factors... Therefore, fewer large windthrows may be identified in this study in northwestern Amazon due to relatively faster recovery.

II.176 Please note in text that provided uncertainties are based on 1 standard deviation

Fixed.

II 43-44: “CAPE determines the speed limit of upward convection.”, which means it could also be less (entrainment), or in fact more if conditions are strongly sheared <https://doi.org/10.1175/JAS-D-20-0103.1> I don't want to get into the question of not considered shear here though. There are bigger problems in this manuscript. However, the sole focus on CAPE for storm intensity should at least be discussed and the exclusion of other possibly important variables mentioned.

We have added explanation on the sole focus on CAPE in the introduction section.

[Line 83-88, Introduction section] Compared to the relatively uniform geographic variability of weak windshear¹¹ over the Amazon region (Fig. S1), CAPE shows a greater spatial variation. Furthermore, the frequency of severe thunderstorm environments increases with warming across the tropics, mainly due to changes in the distribution of CAPE¹³. Therefore, CAPE was selected for this study as the only climate variable for the spatial pattern analysis of windthrows.

Fig. S1 A more uniform spatial pattern of windshear, which is relatively weak over the tropics compared to shear in mid-latitudes¹. Mean monthly windshear over 30 years (1990-2019). Wind shear was calculated as the difference between the horizontal wind vector near the surface and

6km above the surface. The calculation of windshear in this study followed Seeley and Romps 2015² and used ERA5 monthly averaged V-component of wind and U-component of wind on surface pressure and 100 mbar.

References

- Rasmussen, K. L., Prein, A. F., Rasmussen, R. M., Ikeda, K., & Liu, C. (2020). Changes in the convective population and thermodynamic environments in convection-permitting regional climate simulations over the United States. *Climate Dynamics*, 55(1–2), 383–408. <https://doi.org/10.1007/s00382-017-4000-7>
- Garstang, M., Harold L. Massie, J., Halverson, J., Greco, S., & Scala, J. (1994). Amazon Coastal Squall Lines, Part I: structure and Kinematics. *Monthly Weather Review*, 122(4), 608–622.
- Garstang, M., White, S., Shugart, H. H., & Halverson, J. (1998). Convective cloud downdrafts as the cause of large blowdowns in the Amazon rainforest. *Meteorology and Atmospheric Physics*, 67(1–4), 199–212. <https://doi.org/10.1007/BF01277510>
- Negrón-Juárez, R. I., Chambers, J. Q., Guimaraes, G., Zeng, H., Raupp, C. F. M., Marra, D. M., Ribeiro, G. H. P. M., Saatchi, S. S., Nelson, B. W., & Higuchi, N. (2010). Widespread Amazon forest tree mortality from a single cross-basin squall line event. *Geophysical Research Letters*, 37(16), 1–5. <https://doi.org/10.1029/2010GL043733>

REVIEWER COMMENTS

Reviewer #1 (Remarks to the Author):

This is an important study that will contribute meaningfully to the literature. The revisions improved the manuscript, particularly in terms of transparency in the methods. However, I have major concerns about how the threshold for high/low CAPE was defined and how the CAPE-disturbance density relationship was analyzed and projected. Additional details in the methods are also needed for this study to be repeated and clearly interpreted.

Major comments

1. The threshold used to define low-versus-high CAPE (1032 J kg⁻¹) has no physical basis, the underlying projections based on the current binning approach are flawed, and the CAPE-disturbance relationship does not have statistical support. If you include a projection based on the relationship between these two continuous variables (CAPE and disturbance density), then the projection should use the linear relationship between these variables if possible. Projections based on binning are only appropriate when the data cannot be fit by a reasonable linear relationship, but that is unlikely to be the case here. The scatter plot in the previous version indicated that these data would be well-fit by a generalized linear model with Poisson errors and a log-link function, which is a common model structure and should be easy to incorporate (I would also suggest accounting for spatial autocorrelation, which can be done with spatial eigenvectors, random effects of nested spatial bins, or various other approaches). I realize that a previous reviewer suggested dropping the linear model, but this binned approach is much more problematic when projecting future disturbance densities. An additional, and quite meaningful, benefit of including the linear relationship is that it provides statistical support for the claim that disturbance density increases with CAPE across the Amazon. Without this, the main claim of the study rests entirely on the readers' interpretation of visual patterns, which makes this claim unnecessarily weak given the easy options for appropriate analyses.

Similarly, if you retain the threshold approach to defining low and high CAPE regions, then you need to use a mechanistically-based approach to defining this threshold. The current approach identifies the threshold as the highest CAPE value among the lowest-disturbance density cells containing a total of 20% of observed disturbances. This approach is susceptible to strong biases associated with sampling extent and detection efficiency, and it has no mechanistic connection to the influence of CAPE on disturbances. If the authors are keen to identify a threshold, then they could use approaches like breakpoint or segmented regression to test whether a threshold exists. Accordingly, either use a continuous measure of the CAPE-disturbance density relationship or use a statistically robust approach to defining a threshold above which CAPE causes more disturbance.

2. There remain some issues with reproducibility of the methods. Specific issues are outlined below, but it is possible that not all possible issues are identified here. Please carefully review the methods and ensure that future studies could replicate this study based on the description provided in the text.

Minor comments

The "conceptual model" is misnamed and therefore misleading. Consider a different naming approach that represents what is done here (it is neither a model nor is it very conceptual)

Line 56. Remove the "the" before "key"

Line 70. Please clarify what are these 38 observations.

Line 86-87. This sentence says that thunderstorm frequency increases because of increasing CAPE. These variables can be correlated, but the earlier text indicates that we

do not know that CAPE causes more thunderstorms. Consider revising to emphasize the relationship without implying that it is known to be causative

Line 285-286. Please briefly clarify how the spectral mixture analysis identified candidate windthrows. Presumably this procedure identified large areas with less photosynthetic vegetation which were then inspected visually, but this is not clear to a casual reader. It should be clear which aspects of the process were automated and which were manual.

Line 289-290. The response to reviewers and the main text include different claims about how recently these disturbances occurred. This disagreement suggests that the true date is not known and the age classification (more or less than 1 year in the main text, but 2 years in the response to reviewers) is a guess. Please be transparent about what exactly is known about these disturbances and their age.

Line 291-292. "occurred over 1 year from" should be replaced with "occurred more than 1 year before"

Line 292-293. It is not clear from this text how the windthrows were verified with old images. Please describe this process in greater depth so that these methods could be repeated. Additionally, please provide more information about how many disturbances were "old" (and required verification) and how many were "new." It could be valuable to present supplemental figures with the new and old disturbances plotted separately

Line 303-305. Please clarify the difference between being able to identify the year of occurrence (125 windthrows) and having clear remote sensing evidence of the date (38 windthrows). Also clarify whether the 38 windthrows are a subset of the 125.

Line 311-312. This is unclear. If you assumed that the 1012 windthrows occurred from 1990-2019 because the 125 disturbances occurred in this timeframe, then explicitly state that you made this assumption. If the occurrence year of the 1012 windthrows was somehow validated, then provide more detail about this validation procedure.

Line 316. Replace "grids" with "grid"

Line 325 and 327. "Levels" is unclear in both instances. Please clarify the meaning of "levels" and/or use a different word

Line 337. Replace "was" with "were"

Figure 2. The separation of the bins is not well justified and the raw data are not provided. The methods refers to Figures 2a and 2b as histograms. However, this is not correct (these are barplots), and the figures use an unusual binning method wherein the sum of the response variable within each bin is held equal across bins. These data would be more intuitive to interpret if a scatterplot of windthrow density versus CAPE was provided along with the appropriate modeling approach that is suggested above. I suggest replacing Fig. 2a with this continuous relationship and plotting true histograms (or density plots) of CAPE (x-axis) by area (y-axis) for current and future scenarios in Fig. 2b. This would allow the reader to visualize the true relationships in the raw data and it would improve confidence in the results.

Reviewer #2 (Remarks to the Author):

This is a much improved version of the manuscript titled "Climate warming projected to cause a large increase in Amazon wind throw disturbance this century", where the authors sufficiently addressed my concerns regarding validity of their method as the parametric model was removed and the focus is now on a potential area increase of a

CAPE threshold that favours convective storm development rather than discussing storm intensity effects on windthrow. In addition, the methods section has been expanded and provides the needed detail for understanding. I suggest publication of this manuscript after some minor corrections. All line references are for the tracked-changes document:

Abstract: Lacks information on used emission scenario. Change I.22: "[...] by the end of this century under the highest existing emission scenario SSP585.", which does not correspond to the current policies warming trajectory. The abstract should avoid to say "expected warming", which is misleading. This study considers an extreme-case warming.

Title: Should similarly reflect consideration of one extreme emission scenario. Could be "Climate warming projected to cause a large increase in Amazon windthrow disturbance this century under a high emission scenario". Otherwise, title would have to be changed to focus on mechanism relationship rather than a semi-quantification ("large") of future change, e.g. like "Amazon windthrow disturbances are likely to increase with storm frequencies under global warming"

II. 49-52: change to "influences the development_ of further convection_."

II 357-359: "The scaled CMIP6 future CAPE values were comparable with ERA 5 current mean afternoon CAPE values" reads like future CAPE is similar to current CAPE. It would be clearer to say that the scaled CMIP6 future CAPE values are within plausible range compared to the ERA5 current mean afternoon CAPE values.

Fig S.1 caption: It is not obvious by what measure wind shear is "more homogeneous" and what the conclusion from that is. The caption should maybe more precisely state "we find no correspondence between climatological wind shear patterns and windthrow density". It is also not clear what the comparison of mid-latitude wind shear being stronger than in the tropics adds to the argument – wind shear is the determining factor for the development of squall lines, also in the tropics, which the authors specifically mention to be the type of destructive storm causing wind throw. The mid-latitude reference should be removed and the pattern correspondence of CAPE rather than wind shear to windthrow be highlighted.

II385-387: "Moreover, the non-parametric model makes the conservative assumption that the windthrow density does not increase at higher, as-yet unobserved values of mean afternoon CAPE." Also worth to note however that the 'scaling' with relative deltaCAPE (%) rather than absolute deltaCAPE (J/kg), albeit a justified choice given different CAPE definitions between ERA5 & CMIP, is more sensitive to the (often biased) CMIP historical baseline conditions than absolute changes, likely introducing a larger scaled spread (lower/higher min/max CAPE changes) in the deltaCAPE estimates. It is hence the less conservative choice but better reflects the uncertainty range.

LI157: change to "38% of _the_ Amazon region has"

II162: specify "+/- 7%" ?

II162-164: change to "[...] is a strong predictor _of the spatial pattern of large windthrow events_ [...] _that can map windthrow density based on CAPE values._"

II179-180: change to "the _magnitude_ of future CAPE indicates strong instability of _the_ atmosphere [...]". Or say "atmospheric instability"

II180: should say "more frequent storms" rather than "more severe storms", as the study is not looking at effects of increased severity of storms.

L189: windthrow density _is_ relatively _high_

II190: experience _a_ 33%-50% increase

I197: All numbers are round_ed_

I208: in _the_ Amazon region

I209: Amazon_ian_ forests

I211: central Amazon_ia (or _the_ central Amazon)

I213 _the_ central Amazon

I215 _the_ drought year of 2005

I216 increasing _frequency_ (?) of convective storms

I222 _A_ mazon region

I223 "are projected to experience average conditions favourable for intense convective storms" rather than "projected to experience extreme convective storms" - where does the notion of "extreme" come from? Further, "by the end of this century" should be "by the end of this century _under the highest emission scenario SSP585_".

LI 228-230: "The projected significant increase in tree mortality associated with windthrow density in the Amazon by the end of the 21st century has the potential to re-shape tropical forest structure and composition, with sizable impacts on regional carbon balance."

The discussion currently completely ignores the sole focus on the SSP858 scenario of this study. Please be specific and change to "[..] by the end of the 21st century under the SSP858 scenario [..]", and emphasize in the discussion (rather than only in methods section) that this study considers a high-end emission scenario only.

Reviewer #1 (Remarks to the Author):

This is an important study that will contribute meaningfully to the literature. The revisions improved the manuscript, particularly in terms of transparency in the methods. However, I have major concerns about how the threshold for high/low CAPE was defined and how the CAPE-disturbance density relationship was analyzed and projected. Additional details in the methods are also needed for this study to be repeated and clearly interpreted.

Thank you for your continued support in this manuscript. We appreciate your suggestions.

Major comments

1. The threshold used to define low-versus-high CAPE (1032 J kg^{-1}) has no physical basis, the underlying projections based on the current binning approach are flawed, and the CAPE-disturbance relationship does not have statistical support. If you include a projection based on the relationship between these two continuous variables (CAPE and disturbance density), then the projection should use the linear relationship between these variables if possible. Projections based on binning are only appropriate when the data cannot be fit by a reasonable linear relationship, but that is unlikely to be the case here. The scatter plot in the previous version indicated that these data would be well-fit by a generalized linear model with Poisson errors and a log-link function, which is a common model structure and should be easy to incorporate (I would also suggest accounting for spatial autocorrelation, which can be done with spatial eigenvectors, random effects of nested spatial bins, or various other approaches). I realize that a previous reviewer suggested dropping the linear model, but this binned approach is much more problematic when projecting future disturbance densities. An additional, and quite meaningful, benefit of including the linear relationship is that it provides statistical support for the claim that disturbance density increases with CAPE across the Amazon. Without this, the main claim of the study rests entirely on the readers' interpretation of visual patterns, which makes this claim unnecessarily weak given the easy options for appropriate analyses.

Similarly, if you retain the threshold approach to defining low and high CAPE regions, then you need to use a mechanistically-based approach to defining this threshold. The current approach identifies the threshold as the highest CAPE value among the lowest-disturbance density cells containing a total of 20% of observed disturbances. This approach is susceptible to strong biases associated with sampling extent and detection efficiency, and it has no mechanistic connection to the influence of CAPE on disturbances. If the authors are keen to identify a threshold, then they could use approaches like breakpoint or segmented regression to test whether a threshold exists. Accordingly, either use a continuous measure of the CAPE-disturbance density relationship or use a statistically robust approach to defining a threshold above which CAPE causes more disturbance.

Thanks for raising this question. Sorry we didn't make it very clear in the previous version.

We have followed your suggestion and added a segmented regression on aggregated CAPE and windthrow density datasets to further support the threshold (Fig. S4). The segmented regression identified a threshold at $1013.72 \text{ J kg}^{-1}$, which provides statistical support for the threshold (1023 J kg^{-1}) used in this study.

Fig. S4 A segmented regression using aggregated mean afternoon CAPE and windthrow density datasets from Fig. 1b and 1d shows that a threshold exists at CAPE of $1013.72 \text{ J kg}^{-1}$ ($r^2 = 0.39$, $p < 0.05$). The confidence interval of the threshold is $897.5 - 1129.9 \text{ J kg}^{-1}$.

The threshold method is widely used in thunderstorm and CAPE research. For example, damaging thunderstorm environments is defined when $\text{CAPE} \times \text{vertical wind shear}$ exceeds a threshold (Singh et al. 2017, Seeley and Roms 2014). On average, CAPE of 1000 J kg^{-1} is recognized as a threshold for strong to severe storms (National Weather Service, NOAA), and this CAPE threshold is well supported by CAPE theory (Rennó and Ingersoll, 1996) and numerical model (Klemp and Weisman., 1982) and validated by observations of severe storm cases in these studies.

We think this threshold approach is more appropriate than statistical relationship for this study. First, the results focusing on the increased area of storm-favorable environment identified using threshold requires fewer assumptions than directly focusing on the storm frequency (as explained in Line 261-270, Discussion). Therefore, the threshold method is more robust in this way, as suggested by the other reviewer. Second, we have found few windthrows in regions with high CAPE values due to the high cloud coverage. A statistical approach may cause more biases at high CAPE values with only a few data points. The bins of the step function contain same amount of windthrows so that fewer biases are introduced.

We have added statistical and literature support for the CAPE threshold in results and mentioned the incomplete observations due to high cloud coverage.

[Line 154-157, Results] Fig. 2a shows that a CAPE threshold at 1023 J kg^{-1} can be used to define if the environment is favorable for high density of windthrows. A similar threshold is defined by a mechanistically-based approach using the data shown in Fig. 1b and 1d (Fig. S4) and is supported by convection theory, numerical models, and observations in previous studies^{32,33}.

[Line 402-404, Methods] It is worth noting that the future windthrow density produced by models may be underestimated because the windthrow observations within regions with high CAPE were incomplete due to high cloud coverage.

Singh, M. S., Kuang, Z., Maloney, E. D., Hannah, W. M., & Wolding, B. O. (2017). Increasing potential for intense tropical and subtropical thunderstorms under global warming. *Proceedings of the National Academy of Sciences of the United States of America*, 114(44), 11657–11662. <https://doi.org/10.1073/pnas.1707603114>

Seeley, J. T., & Romps, D. M. (2015). The effect of global warming on severe thunderstorms in the United States. *Journal of Climate*, 28(6), 2443–2458. <https://doi.org/10.1175/JCLI-D-14-00382.1>

National Weather Service, NOAA. Convective Parameters – Cape. November 3, 2022. <https://www.weather.gov/fwd/convectiveparameterscape>

Rennó, N., & Ingersoll, A. (1996). Natural Convection as a Heat Engine: A Theory for CAPE. *Journal of the Atmospheric Sciences*, 53(4), 572–585.

Klemp, M. L., & Weisman, J. B. (1982). The dependence of numerically simulated convective storms on vertical wind shear and buoyancy. *Monthly Weather Review*, 110, 504–520.

2. There remain some issues with reproducibility of the methods. Specific issues are outlined below, but it is possible that not all possible issues are identified here. Please carefully review the methods and ensure that future studies could replicate this study based on the description provided in the text.

Thanks for your suggestions, we have carefully reviewed the methods and added more details in methods to make sure the approach can be reproducible.

Minor comments

The “conceptual model” is misnamed and therefore misleading. Consider a different naming approach that represents what is done here (it is neither a model nor is it very conceptual)

Thanks for the suggestion. We have changed “the conceptual model” to “empirical relationship” when describing Fig. 2a. We retain the use of “model based on the relationship between CAPE and windthrow density” when referring to the future projections of windthrow density. The code of the model for future windthrow projection can be found in code availability section.

[Line 146, Results] A relationship maps CAPE to windthrow density

[Line 147, Results] We found an empirical relationship using mean afternoon CAPE to reproduce the ...

[Line 163, Results] The relationship shows that CAPE is a strong predictor of the spatial pattern ..

[Line 168-170, Results] We developed a model based on the CAPE and windthrow density relationship to assess how climate change will affect CAPE and then affect the density of future large windthrows in the Amazon region.

Line 56. Remove the “the” before “key”

Fixed

Line 70. Please clarify what are these 38 observations.

We think “38 observations” is unnecessary and delete it. It originally refers to windthrow case studies shown in reference No.24.

No. 24 Feng, Y., Negrón-Juárez, R. I. & Chambers, J. Q. Case studies of Forest Windthrows and Mesoscale Convective Systems in Amazon forest. *Manuscr. Submitt. Publ.* (2022).

Line 86-87. This sentence says that thunderstorm frequency increases because of increasing CAPE. These variables can be correlated, but the earlier text indicates that we do not know that CAPE causes more thunderstorms. Consider revising to emphasize the relationship without implying that it is known to be causative

Thanks for the question. Here we say that “the frequency of severe thunderstorm environments increases with CAPE...” rather than “the thunderstorm frequency increase”. The severe thunderstorm environment is different from the storm frequency. Therefore, we imply CAPE causes more favorable environments for thunderstorms rather than more thunderstorms. We think it is clear stated in the sentence.

Line 285-286. Please briefly clarify how the spectral mixture analysis identified candidate windthrows. Presumably this procedure identified large areas with less photosynthetic vegetation which were then inspected visually, but this is not clear to a casual reader. It should be clear which aspects of the process were automated and which were manual.

Thanks for the suggestions. We have added more explanations on spectral mixture analysis in methods.

[Line 291-295, Methods] Each pixel was unmixed to fractions of image-derived endmembers, including green vegetation (GV), non-photosynthetic vegetation (NPV), and shade. GV and NPV fraction images were normalized without shade and then used to identify windthrows.

Windthrows were identified manually as large fan-shape areas with high NPV fraction.

Line 289-290. The response to reviewers and the main text include different claims about how recently these disturbances occurred. This disagreement suggests that the true date is not known and the age classification (more or less than 1 year in the main text, but 2 years in the response to reviewers) is a guess. Please be transparent about what exactly is known about these disturbances and their age.

Sorry for the confusion. It should be 1 year as stated in the main text. It was an error in the response to reviewers letter.

Line 291-292. “occurred over 1 year from” should be replaced with “occurred more than 1 year before”

Fixed

Line 292-293. It is not clear from this text how the windthrows were verified with old images. Please describe this process in greater depth so that these methods could be repeated.

Additionally, please provide more information about how many disturbances were “old” (and required verification) and how many were “new.” It could be valuable to present supplemental figures with the new and old disturbances plotted separately

Thanks for the suggestions. We have added more explanations on the process of windthrow verification using historical Landsat images. We have also added Fig. S7c to illustrate the process. Figures of “old” and “new” windthrows were provided in Fig. S7a and b.

[Line 299-305, Methods] “New” windthrows that occurred within 1 year were spectrally more visible based on their clear fan-shape^{5,10} (diverging from a central area with small pixels scattered at the tail) and their relatively distinguishable reddish colors (Fig. S7a, due to high reflectance in shortwave infrared band from woody biomass), while “old” windthrows (Fig. S7b) occurred more than 1 year before the identification were displayed in bright green colors (due to reflectance in near infrared band from the pioneer species). “Old” windthrows account for around 80% of total identified windthrows, and they were verified using historical Landsat images that can go as far as 1984 (when Landsat 5 was launched). “Old” windthrows were validated once they were found with clear shape and more distinguish color on the historical Landsat images (Fig. S7c).

a

“New” windthrow:
centered at -74.25634° , -3.75625° ,
displayed in Landsat composite,
displayed in false color (red: SWIR
band, green: NIR band, blue: red
band) from 2018-2019, the
windthrow occurred in 07/30/2017

b

“Old” windthrow:
centered at -59.12147° , -2.81343° ,
displayed in Landsat composite,
displayed in false color (red: SWIR band,
green: NIR band, blue: red band) from
2018-2019, the windthrow occurred in
10/16/2000

c

The same “old” windthrow as Fig. S7 (b) displayed in Landsat 5 image from 11/29/2000. The shape and color of the windthrow is more visible and this image can be used to validate the existence of this “old” windthrow.

Fig. S7 (a) “New” Windthrow that occurred within 2 years of identification displayed in satellite images from 2018-2019. The reddish color indicated that the windthrow was relatively fresh with lots of dead trees with high reflectance in SWIR; (b) “Old” windthrow that occurred nearly 20 years ago from identification displayed in satellite images from 2018-2019; (c) “Old” windthrow that occurred nearly 20 years ago from identification displayed in satellite images from 2000

Line 303-305. Please clarify the difference between being able to identify the year of occurrence (125 windthrows) and having clear remote sensing evidence of the date (38 windthrows). Also clarify whether the 38 windthrows are a subset of the 125.

Thanks for the suggestions. Another of our study (reference No.24) which is currently under review has a detailed documentation of how the year and date of windthrows are identified. We have added more details and cited the paper.

[Line 317-319, Methods] ...and 38 windthrows from these 125 windthrows had clear remote sensing A detailed process of identifying the occurrence year and date of windthrow can be found in a study on windthrow cases by Feng et al. under review²⁴.

Line 311-312. This is unclear. If you assumed that the 1012 windthrows occurred from 1990-2019 because the 125 disturbances occurred in this timeframe, then explicitly state that you made this assumption. If the occurrence year of the 1012 windthrows was somehow validated, then provide more detail about this validation procedure.

Thanks for the question. We assumed that the 1012 windthrows occurred from 1990-2019 based on the reasons we explained in line 319-323, and then we validated our assumption using the occurrence year of 125 windthrow cases.

We have added more explanations to make the process clearer.

[Line 324-326, Methods] Based on the recovery time, we estimated that these 1012 windthrows most likely occurred within 30 years (between 1990-2019), and the estimated occurrence period was validated using the range of the occurrence year (1986-2019) of 125 windthrow cases.

Line 316. Replace “grids” with “grid”
Fixed

Line 325 and 327. “Levels” is unclear in both instances. Please clarify the meaning of “levels” and/or use a different word

Thanks for your suggestion. “Levels” refers to pressure levels we have added “pressure” to make it clearer.

Line 337. Replace “was” with “were”
Fixed.

Figure 2. The separation of the bins is not well justified and the raw data are not provided. The methods refers to Figures 2a and 2b as histograms. However, this is not correct (these are barplots), and the figures use an unusual binning method wherein the sum of the response variable within each bin is held equal across bins. These data would be more intuitive to interpret if a scatterplot of windthrow density versus CAPE was provided along with the appropriate modeling approach that is suggested above. I suggest replacing Fig. 2a with this continuous relationship and plotting true histograms (or density plots) of CAPE (x-axis) by area (y-axis) for current and future scenarios in Fig. 2b. This would allow the reader to visualize the true relationships in the raw data and it would improve confidence in the results.

We appreciate your suggestions on the scatter plot and raw datasets. The raw data and the code to produce Fig. 2a can be found data and code availability section. We have a detailed paragraph (line 386-400) describing the methods to generate step function shown in Fig.2a. As explained in the response to the first question, we have also added the scatter plot in the supplementary material.

[Line 386-400, Methods] We developed a model based on the relationship between satellite-derived windthrow density and mean afternoon CAPE from the ERA 5 reanalysis over 1990-2019. The non-parametric model provides a look-up table of windthrow density as a function of CAPE within the range of observations. Counts of observed windthrow events and Amazon’s area were separately binned by CAPE using the same bins, producing two histograms of CAPE. The ratio of the former to the latter gives the density of windthrow events (windthrow events per area) as a function of CAPE. To avoid noise at the tails of the histograms, the six CAPE bins were chosen such that each bin would have about the same number of windthrow events (either 168 or 169). The total number of windthrow events is given by the sum over bins of the product of windthrow density and area. The minimum and maximum of current ERA 5 mean afternoon CAPE was 42 and 1549. The minimum CAPE value of the first bin was extended to 0 and the maximum CAPE value of the last bin extended to infinity under the assumption that the windthrow density is similar for neighboring values. Since the windthrow density is roughly a step function in CAPE (see Fig. 2a), it is the increase in the area with high CAPE that then leads to an increase in the number of windthrow events.

Reviewer #2 (Remarks to the Author):

This is a much improved version of the manuscript titled “Climate warming projected to cause a large increase in Amazon wind throw disturbance this century”, where the authors sufficiently addressed my concerns regarding validity of their method as the parametric model was removed and the focus is now on a potential area increase of a CAPE threshold that favours convective storm development rather than discussing storm intensity effects on windthrow. In addition, the methods section has been expanded and provides the needed detail for understanding. I suggest publication of this manuscript after some minor corrections. All line references are for the tracked-changes document:

Thank you for your continued support and all the suggestions.

Abstract: Lacks information on used emission scenario. Change 1.22: “[..] by the end of this century under the highest existing emission scenario SSP585.”, which does not correspond to the current policies warming trajectory. The abstract should avoid to say “expected warming”, which is misleading. This study considers an extreme-case warming.

Thanks for the suggestions. We have added the emission scenario based on your suggestions. [Line 21-22, Abstract]... by the end of this century under the high-emission scenario (SSP 585).

Title: Should similarly reflect consideration of one extreme emission scenario. Could be “Climate warming projected to cause a large increase in Amazon windthrow disturbance this century under a high emission scenario”. Otherwise, title would have to be changed to focus on mechanism relationship rather than a semi-quantification (“large”) of future change, e.g. like “Amazon windthrow disturbances are likely to increase with storm frequencies under global warming”

Thanks for the suggestions. We have adopted your second suggestion on the title and changed it to “Amazon windthrow disturbances are likely to increase with storm frequency under global warming”.

ll. 49-52: change to “influences the development_ of further convection_.”
Fixed.

ll 357-359: “The scaled CMIP6 future CAPE values were comparable with ERA 5 current mean afternoon CAPE values” reads like future CAPE is similar to current CAPE. It would be clearer to say that the scaled CMIP6 future CAPE values are within plausible range compared to the ERA5 current mean afternoon CAPE values.

Fixed. Thanks for pointing it out.

[Line 373, Methods] The scaled CMIP6 future CAPE values were within plausible range compared to the ERA 5 current mean afternoon CAPE values...

Fig S.1 caption: It is not obvious by what measure wind shear is “more homogeneous” and what the conclusion from that is. The caption should maybe more precisely state “we find no

correspondence between climatological wind shear patterns and windthrow density”. It is also not clear what the comparison of mid-latitude wind shear being stronger than in the tropics adds to the argument – wind shear is the determining factor for the development of squall lines, also in the tropics, which the authors specifically mention to be the type of destructive storm causing wind throw. The mid-latitude reference should be removed and the pattern correspondence of CAPE rather than wind shear to windthrow be highlighted.

Thanks for your suggestions. We have followed your suggestions and made related changes. [Line 3-4 in supplementary material, Fig. S1 caption] Mean monthly windshear over 30 years (1990-2019). We find no correspondence between climatological wind shear patterns and windthrow density…]

l1385-387: “Moreover, the non-parametric model makes the conservative assumption that the windthrow density does not increase at higher, as-yet unobserved values of mean afternoon CAPE.” Also worth to note however that the ‘scaling’ with relative deltaCAPE (%) rather than absolute deltaCAPE (J/kg), albeit a justified choice given different CAPE definitions between ERA5 & CMIP, is more sensitive to the (often biased) CMIP historical baseline conditions than absolute changes, likely introducing a larger scaled spread (lower/higher min/max CAPE changes) in the deltaCAPE estimates. It is hence the less conservative choice but better reflects the uncertainty range.

Thanks for pointing out this point. We have added it to the methods part. [Line 375-378, Methods] However, it is worth noting that the scaling with relative changes in delta CAPE (%) is more sensitive to CMIP historical baseline conditions than absolute changes of CAPE ($J\ kg^{-1}$), which will likely introduce a larger scaled spread (min/max CAPE changes).

L1157: change to “38% of the Amazon region has”
Fixed.

l1162: specify “+/- 7%” ?

Thanks for the suggestions. The standard deviation (SD) should always be a positive number based on the equation to calculate SD. We have replaced 7% (SD/mean fraction) with the true value of 1 SD (0.2), which we think is a clearer representation. [Line 163, Results] The mean uncertainty (SD) of these values, estimated via bootstrapping, is 0.2.

l1162-164: change to “[.] is a strong predictor of the spatial pattern of large windthrow events [.] that can map windthrow density based on CAPE values.”
Fixed.

l1179-180: change to “the magnitude of future CAPE indicates strong instability of the atmosphere [.]”. Or say “atmospheric instability”
Fixed.

l1180: should say “more frequent storms” rather than “more severe storms”, as the study is not looking at effects of increased severity of storms.
Fixed.

L1189: windthrow density is relatively high

Fixed.

1190: experience _a_ 33%-50% increase

Fixed.

1197: All numbers are round _ed_

Fixed.

1208: in _the_ Amazon region

Fixed.

1209: Amazon _ian_ forests

Fixed.

1211: central Amazon _ia_ (or _the_ central Amazon)

Fixed.

1213 _the_ central Amazon

Fixed.

1215 _the_ drought year of 2005

Fixed.

1216 increasing _frequency_ (?) of convective storms

Fixed.

1222 _A_ mazon region

Fixed.

1223 “are projected to experience average conditions favourable for intense convective storms” rather than “projected to experience extreme convective storms” - where does the notion of “extreme” come from? Further, “by the end of this century” should be “by the end of this century under the highest emission scenario SSP585”.

We have adopted your suggestions and changed the texts.

[Line 226-227, Discussion]... experience average conditions favorable for intense convective storms with a large increase in windthrow events by the end of this century under the highest emission scenario SSP585.

L1 228-230: “The projected significant increase in tree mortality associated with windthrow density in the Amazon by the end of the 21st century has the potential to re-shape tropical forest structure and composition, with sizable impacts on regional carbon balance.” The discussion currently completely ignores the sole focus on the SSP858 scenario of this study. Please be specific and change to “[..] by the end of the 21st century under the SSP858 scenario [..]”, and emphasize in the discussion (rather than only in methods section) that this study considers a high-end emission scenario only.

We have followed your suggestions and have emphasized it in the results and discussion section. [Line 177-178, Results] Table 1 lists the fractional changes ... under the highest emission scenario SSP585.

[Line 195, Table 1 caption] Table 1 Future changes predicted by ESMs under the highest emission scenario SSP585.

[Line 277, Discussion] The results in this paper generated using ESM outputs ... by the end of this century under the highest emission scenario SSP585.

REVIEWERS' COMMENTS

Reviewer #1 (Remarks to the Author):

The authors have done an excellent job addressing the comments, particularly in terms of improve clarity in the methods. I only have two minor additional comments related to new text in the most recent version of the paper. Great work on a very interesting paper!

Minor comments

Line 315-326. The revisions to this paragraph produced an apparent contradiction that should be addressed. The first part of this paragraph states that 125 windthrow events were examined and only 38 could be dated, whereas the last sentence states that 125 windthrow events were dated. Please clarify this apparent disagreement in the text. Additionally, clarify why the authors are confident in the date range when it seems that the vast majority of windthrow events could not be dated (namely, state why you are confident that that the disturbances did not occur before 1986 while most disturbances could not be dated)

Line 398-400. This statement is not true and should be corrected. The windthrow density does not resemble a step-function of CAPE. It only appears that way in Figure 2A because the data are grouped so that the same number of windthrows are in each unequally-sized bin. Fig. S4 shows the empirical patterns, which exhibit a clear linear trend. Elsewhere in the paper it states that trends in windthrow density are not projected for high CAPE regions because data are sparse at high CAPE values and those projections would be unreliable, which is justifiable. However, there is no evidence in this study suggesting that increases in CAPE in the high CAPE areas will not increase the windthrow density in those regions.

Reviewer #1 (Remarks to the Author):

The authors have done an excellent job addressing the comments, particularly in terms of improve clarity in the methods. I only have two minor additional comments related to new text in the most recent version of the paper. Great work on a very interesting paper!

Thank you for taking the necessary time and effort to review the manuscript. All your comments and suggestions helped us improve the quality of the manuscript.

Minor comments

Line 315-326. The revisions to this paragraph produced an apparent contradiction that should be addressed. The first part of this paragraph states that 125 windthrow events were examined and only 38 could be dated, whereas the last sentence states that 125 windthrow events were dated. Please clarify this apparent disagreement in the text. Additionally, clarify why the authors are confident in the date range when it seems that the vast majority of windthrow events could not be dated (namely, state why you are confident that that the disturbances did not occur before 1986 while most disturbances could not be dated)

Thanks for the questions. We would like to explain that it is the year of 125 windthrows were identified, while the date of 38 windthrows were identified. It is relatively easier to identify the year rather than date of windthrows. We have added explanations to avoid misunderstandings.

[Line 323-324, Methods] It is difficult to get the accurate year and date of occurrence of all identified windthrows.

Additionally, we have added more explanations to clarify why we are confident in the windthrow occurrence date range.

[Line 327-329, Methods] Based on the recovery time (20-40 years) and the time of windthrow identification (2018-2019), we estimated that these 1012 windthrows most likely occurred within 30 years...

Line 398-400. This statement is not true and should be corrected. The windthrow density does not resemble a step-function of CAPE. It only appears that way in Figure 2A because the data are grouped so that the same number of windthrows are in each unequally-sized bin. Fig. S4 shows the empirical patterns, which exhibit a clear linear trend. Elsewhere in the paper it states that trends in windthrow density are not projected for high CAPE regions because data are sparse at high CAPE values and those projections would be unreliable, which is justifiable. However, there is no evidence in this study suggesting that increases in CAPE in the high CAPE areas will not increase the windthrow density in those regions.

Thanks for the comments. We have corrected the statement based on your comments.

[Line 403-405, Methods] Based on the windthrow density and CAPE relationship used in the model, it is the increase in the area with high CAPE that then leads to an increase in the number of windthrow events.

In addition, we made clarifications in line 408-410 on our conservative assumption on windthrow density within high CAPE regions.

[Line 409-411, Methods] Moreover, the non-parametric model makes the conservative assumption that the windthrow density does not increase at higher, as-yet unobserved values of mean afternoon CAPE.